



# Quality controls, bias, and seasonality of $CO_2$ columns in the Boreal Forest with OCO-2, TCCON, and EM27/SUN measurements

Nicole Jacobs[1], William R. Simpson[1], Debra Wunch[2], Christopher W. O'Dell[3], Gregory B. Osterman[4], Frank Hase[5], Thomas Blumenstock[5], Qiansi Tu[5], Matthias Frey[5,6], Manvendra K. Dubey[7], Harrison A. Parker[7,8], Rigel Kivi[9], and Pauli Heikkinen[9]

[1]Department of Chemistry and the Geophysical Institute, University of Alaska Fairbanks, Fairbanks, AK, USA
[2]Department of Physics, University of Toronto, Toronto, Canada
[3]Cooperative Institute for Research in the Atmosphere, Colorado State University, Fort Collins, CO, USA
[4]Jet Propulsion Laboratory, California Institute of Technology, Pasadena, CA, USA
[5]Karlsruhe Institute of Technology, Institute of Meteorology and Climate Research, Karlsruhe, Germany
[6]National Institute for Environmental Studies, Tsukuba, Japan
[7]Earth and Environmental Sciences, Los Alamos National Laboratory, Los Alamos, NM, USA
[8]California Institute of Technology, Pasadena, CA, USA
[9]Finnish Meteorological Institute, Sodankylä, Finland

**Correspondence:** William Simpson (wrsimpson@alaska.edu)

**Abstract.** Seasonal $CO_2$ exchange in the Boreal Forest plays an important role in the global carbon budget and in driving interannual variability in seasonal cycles of atmospheric $CO_2$. Satellite-based observations from polar orbiting satellites like the Orbiting Carbon Observatory-2 (OCO-2) offer an opportunity to characterize Boreal Forest seasonal cycles across longitudes with a spatially and temporally rich dataset, but data quality controls and biases still require vetting at high latitudes. With

the objective of improving data availability at northern, terrestrial high latitudes, this study evaluates quality control methods and biases of OCO-2 retrievals of atmospheric column-averaged dry-air mole fractions of $CO_2$ ($X_{CO_2}$) in Boreal Forest regions. In addition to the standard quality control filters recommended for ACOS B8 (B8 QC) and ACOS B9 (B9 QC) OCO-2 retrievals, a third set of quality control filters were specifically tailored to Boreal Forest observations (Boreal QC) with the goal of increasing data availability at high latitudes without sacrificing data quality. Ground-based reference measurements

of $X_{CO_2}$ include observations from two sites in the Total Carbon Column Observing Network (TCCON) at East Trout Lake, Saskatchewan, Canada and Sodankylä, Finland. OCO-2 retrievals were also compared to ground-based observations from two Bruker EM27/SUN FTS at Fairbanks, Alaska, United States. EM27/SUN spectrometers that were deployed in Fairbanks were carefully monitored for instrument performance and were bias corrected to TCCON using observations at the Caltech TCCON site. The B9 QC were found to pass approximately twice as many OCO-2 retrievals over land north of $50°$N than the B8

QC, and the Boreal QC were found to pass approximately twice as many retrievals in May, August, and September as the B9 QC. While Boreal QC results in a substantial increase in passable retrievals this is accompanied by increases in the standard deviations in biases at Boreal Forest sites from $\sim 1.4$ ppm with B9 QC to $\sim 1.6$ ppm with Boreal QC. Total average biases for coincident OCO-2 retrievals at the three sites considered did not consistently increase or decrease with different QC methods, and instead responses to changes in QC varied according to site and satellite viewing geometries. Regardless of the quality





control method used, seasonal variability in biases was observed, and this variability was more pronounced at the TCCON sites than when comparing to EM27/SUN observations in Fairbanks. Monthly average biases generally varied between -1 ppm and +1 ppm at the three sites considered, with more negative biases in spring (MAM) and autumn (SO), but more positive biases in summer months (JJA). Monthly standard deviations in biases ranged from approximately 1.0 ppm to 2.0 ppm and do

5   not exhibit strong seasonal dependence apart from exceptionally high standard deviation observed with all three QC methods at Sodankylä in June. There was no evidence found to suggest that seasonal variability in bias is a direct result of airmass dependence in ground-based retrievals or of proximity bias from coincidence criteria, but there were a number of retrieval parameters used as quality control filters that exhibit seasonality and could contribute to seasonal dependence in OCO-2 bias. Furthermore, it was found that OCO-2 retrievals of $X_{CO2}$ without the standard OCO-2 bias correction exhibit almost no

10  perceptible seasonal dependence in average monthly bias at these Boreal Forest sites, suggesting that seasonal variability in bias is introduced by the bias correction. Overall, we found that modified quality controls can allow for significant increases in passable OCO-2 retrievals with only marginal compromises in data quality, but seasonal dependence in biases still warrants further exploration.



# 1 Introduction

The Boreal Forest or Taiga Biome is the largest terrestrial biome on Earth, it includes the sub-Arctic regions of Europe, Asia, and North America between 50°N and 70°N latitudes, it represents an important and influential component of the global carbon cycle, and it is a principle driver of the atmospheric carbon dioxide ($CO_2$) seasonal cycle. Accurate accounting of seasonal $CO_2$ exchange in Boreal Forest regions is an essential component in quantifying the global carbon budget and predicting future climate scenarios (Tans et al., 1990; Pan et al., 2011; Graven et al., 2013; Barlow et al., 2015; Bradshaw and Warkentin, 2015; Gauthier et al., 2015). Studies by Graven et al. (2013) and Barlow et al. (2015) used a combination of atmospheric modeling, aircraft observations, and a network of ground based in-situ observations to investigate seasonal carbon exchange in the Boreal Forest. Both studies found that the Boreal Forest plays an important role in global atmospheric $CO_2$ concentrations, significantly influencing in-situ observations of $CO_2$ in the tropics (Mauna Loa). Multiple studies have shown there is a latitude-dependent trend in the seasonal amplitude of atmospheric $CO_2$ with increased seasonal uptake of $CO_2$ in Boreal Forest regions (Graven et al., 2013; Wunch et al., 2013; Barlow et al., 2015; Lindqvist et al., 2015). Furthermore, the studies by Graven et al. (2013) and Barlow et al. (2015) found that the trend in seasonal cycle amplitudes of $CO_2$ with respect to latitude nearly doubled between 1960 and 2011, suggesting that seasonal changes in the Boreal Forest are growing at an accelerated rate relative to lower latitude regions. While some studies have reported rapid changes in seasonal carbon exchange in the Boreal Forest and proposed that this is a dominant driver in the global carbon budget, another study by Barnes et al. (2016) suggests that it is actually the temperate forest between 30°N and 50°N that is the dominant driver in the global carbon budget. It remains difficult to reconcile conflicting claims about contributions to the global carbon budget without a spatially and temporally rich set of measurements for high latitude regions, and data availability in the Boreal Forest remains a major impediment to accurately quantifying uptake in the world's largest terrestrial biome (Pan et al., 2011; Barlow et al., 2015; Euskirchen et al., 2017). Therefore, methods of expanding observational coverage through improved satellite observations at high latitudes are essential for clarifying our understanding of global $CO_2$ exchange.

Satellite-based observations of atmospheric $CO_2$ columns offer a more holistic view of global $CO_2$ dynamics by expanding spatial coverage. NASA's Orbiting Carbon Observatory 2 (OCO-2) was launched in July 2014 with $CO_2$ column retrievals available from September 2014 to present (OCO-2 Science Team/Michael Gunson, Annmarie Eldering, 2018). Satellite-based observations from OCO-2 consist of solar reflectance in three spectral windows centered at 0.76 $\mu$m, 1.61 $\mu$m, and 2.06 $\mu$m, and referred to as the $O_2$ A band, weak $CO_2$ band, and strong $CO_2$ band, respectively. The ACOS full-physics retrieval algorithm (currently on version 9 or "ACOS B9") described by O'Dell et al. (2012, 2018) and Connor et al. (2008) fits absorption features in these windows and incorporates additional meteorology and model data to retrieve column-averaged dry air mole fractions of atmospheric $CO_2$ ($X_{CO2}$) along with a variety of other parameters, such as aerosol optical depth, surface albedo, surface pressure, and total column water vapor. A number of parameters in the full physics retrievals are used to designate thresholds for post-processing quality control filters. OCO-2 is polar orbiting, so overpasses are more frequent at high latitudes than



mid-latitudes, presenting a valuable opportunity to amass an extensive archive of $CO_2$ observations over the Boreal Forest. However, before OCO-2 can be used to evaluate $CO_2$ seasonality for the Boreal Forest, these data need to be validated in high latitude regions. Quality control filters implemented in previous versions of the ACOS algorithm, like version 8 (ACOS B8) discussed by O'Dell et al. (2018), removed the majority of high latitude observations, and as a result OCO-2 high latitude data

have been underutilized. Validation of OCO-2 satellite-based retrievals at high latitudes has also been limited by the relatively few dedicated ground-based monitoring sites at high latitudes (Wunch et al., 2017b).

In addition to the limited availability of ground-based validation data, there are a number of other challenges to passive satellite measurements at high latitudes. The sun stays low in the sky at high latitudes, even in summer when the sun travels a long azimuthal path it does not reach the same solar elevations as at lower latitudes. Low solar elevation corresponds to a

high solar zenith angle (sza) and high airmass, meaning that sunlight travels a greater distance through the atmosphere before reaching the instrument. High airmasses can cause absorption spectra to become saturated at line center, making column retrievals more sensitive to the line wings and thus the line shape of the absorption line. Spectroscopic uncertainties tend to be exacerbated at higher airmasses, and the relative impacts of radiative transfer effects from atmospheric aerosols on satellite retrievals are also increased at high airmass. In particular, slant-path aerosol optical depths (aod) are larger and scattering angles

are smaller, which increases the fractional contribution of aerosol scatter to the total radiance detected by the satellite. Airmass dependence in passive column measurements continues to be an area of ongoing research in retrieval algorithms (Wunch et al., 2015), and high airmass in winter is one of primary reasons for halting observations at high latitude sites in November through February. Aside from the sunlight, climates and ecosystems at high latitudes are highly seasonally dependent, so there are a number of seasonal parameters that may produce time-dependent biases at high latitudes if they are not handled properly

in retrieval algorithms. In particular, Wu et al. (2018) noted time dependent biases at Sodankylä with the RemoTeC/OCO-2 retrieval algorithm. Wunch et al. (2017b) suggests that there are not enough passable retrievals from ACOS B7 to identify seasonal bias at high latitudes. Snow and ice covered surfaces are known to introduce extensive challenges in passive retrievals of $X_{CO2}$ due to low surface albedo in the weak (1.61 $\mu$m) and strong (2.06 $\mu$m) $CO_2$ bands used by OCO-2 (Wiscombe and Warren, 1980), and reflection anisotropy effects can further complicate retrievals over snow (Boesch et al., 2011; Crisp et al.,

2012). Because snow cover is also seasonal and follows the solar cycle, it may be difficult to isolate causes of seasonal bias at high latitudes. As a result a certain amount of seasonal dependence may be inevitable, but we still endeavor to minimize it with careful attention to quality control methods.

Ground-based column measurements from solar-viewing spectrometers complement passive satellite observations because both use infrared absorption spectroscopy, with the sun as radiation source, and observe a full atmospheric column abundance.

The total carbon column observing network (TCCON) is a ground-based network that uses solar-viewing, high-resolution infrared spectrometers to retrieve $X_{CO2}$ (Wunch et al., 2011a). TCCON is the reference measurement for OCO-2 and is the primary source of validation data. In addition to comparing OCO-2 to TCCON, this paper compares OCO-2 observations to ground-based observations from an EM27/SUN Fourier transform infrared spectrometer (EM27/SUN FTS) operated in Fairbanks, Alaska. The EM27/SUN FTS was developed by the Karlsruhe Institute of Technology (KIT) in collaboration with

Bruker Optics (Gisi et al., 2012; Hase et al., 2016) to be a compact, mobile solar-viewing FTS designed for field deployment.



The EM27/SUN spectrometers have a resolution of 0.5 cm$^{-1}$, similar to that of OCO-2 with $\sim$0.3 cm$^{-1}$ resolution, while the Bruker IFS 125HR used by TCCON has a much higher resolution of $\sim 0.02$ cm$^{-1}$. All three instruments record a solar infrared spectrum that can be used to retrieve $X_{CO2}$. Several recent studies have compared EM27/SUN observations to TCCON (Hedelius et al., 2016, 2017; Velazco et al., 2018; Frey et al., 2019). This paper uses similar retrieval methods for EM27/SUN

retrievals of $X_{CO2}$ as Hedelius et al. (2016), Hedelius et al. (2017), and Velazco et al. (2018) by implementing the GGG2014 retrieval algorithm coupled with the EM27/SUN GGG interferogram processing suite (EGI) (Hedelius and Wennberg, 2017). Hedelius et al. (2016) observed a 0.03% $\pm$0.08% ($\sim 0.12 \pm 0.32$ ppm) offset when comparing four EM27/SUN spectrometers to co-located observations at the Caltech TCCON site. Hedelius et al. (2017) found some EM27/SUN biases to TCCON as large as 0.14% ($\sim 0.56$ ppm), but also found statistically significant variability amongst TCCON sites up to 0.3 ppm, suggesting that

the site-to-site biases amongst TCCON sites may be of similar size to biases between EM27/SUN FTS and TCCON observed by Hedelius et al. (2016). Velazco et al. (2018) found an average offset to TCCON of approximately 0.46% ($\sim 1.84$ ppm) when comparing two years of co-located observations between an EM27/SUN FTS and the TCCON site at the University of Wollongong. While some of these biases are large enough to produce significantly different results when choosing the EM27/SUN FTS or TCCON for ground-based validation of satellite-based $X_{CO2}$ retrievals, these biases tend to be systematic

in nature and can be corrected to achieve acceptable agreement with TCCON through regular calibration measurements. The two EM27/SUN FTS used in Fairbanks were calibrated against the Caltech TCCON and bias corrections were implemented to ensure that both EM27/SUN and TCCON observations are comparable sources of validation data for OCO-2 (see supplemental materials section 1 for details on EM27/SUN instrument comparisons). The EM27/SUNs were also aligned and serviced at the KIT, during which time they were compared to measurements from the Karlsruhe TCCON.

The objective of this study is to explore ways of defining quality control criteria for OCO-2 in high latitude regions, as to maximize spatial and temporal coverage over the Boreal Forest while maintaining acceptable agreement with ground sites. It is also essential that biases in OCO-2 retrieved $X_{CO2}$ be carefully evaluated under different quality control filtering regimes and in the context of high latitude seasonality studies. To this end, we first define retrieval, quality control, and aggregation methods for ground-based measurements that are reasonably equivalent for EM27/SUN or TCCON observations. We verify

that EM27/SUN retrievals of $X_{CO2}$ in Fairbanks are interchangeable with TCCON through comparisons with the Caltech TCCON, which are used to rescale EM27/SUN observations to the TCCON trace-gas scale. Then, we turn to the problem of data paucity in OCO-2 at high northern latitudes during spring and autumn, and investigate the quality control filters applied to those data. We subsequently suggest new quality control filters for Boreal Forest regions that substantially increase OCO-2 high latitude data throughput, and we evaluate the consequences of applying different sets of quality control filters. Finally, we

discuss observed seasonality in OCO-2 biases in the Boreal Forest and explore some retrieval parameters that may contribute to seasonality in bias.





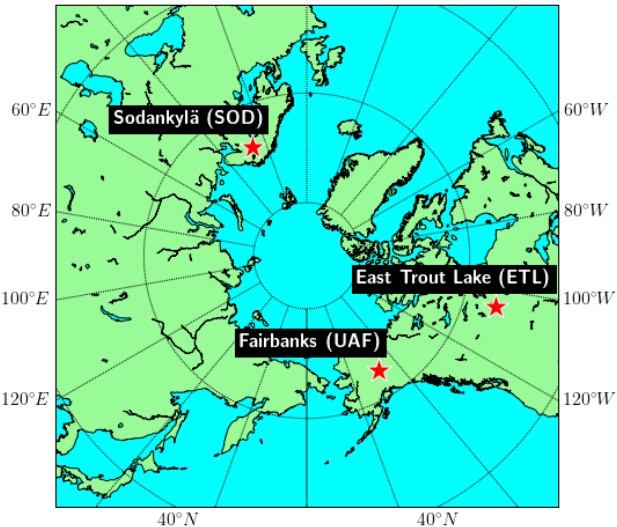

**Figure 1.** Circumpolar map show locations of Boreal Forest sites included in this study.

## 2 Sites and data sources

Ground-based column measurements were collected at three sites in the Boreal Forest including two TCCON sites at East Trout Lake, Saskatchewan, Canada (54.354°N, 104.987°W; "ETL"; Wunch et al. (2017a)) and Sodankylä, Finland (67.367°N, 26.631°E; "SOD"; Kivi et al. (2017)), as well as long-term measurements in Fairbanks, Alaska, U.S. (65.859°N, 147.85°W;

"UAF") using two Bruker EM27/SUN spectrometers (see Fig. 1). Observations at the Sodankylä TCCON site began in 2009, and span the full period of OCO-2 observations considered in this analysis from September 2014 to November 2018 with the exception of winter months (Kivi and Heikkinen, 2016; Kivi et al., 2017). At East Trout Lake observations began in October 2016, and because this site is further south, these measurements are nearly year round (Wunch et al., 2017a). In Fairbanks, the LANL EM27 (owned by Los Alamos National Laboratory) was operated August-October 2016 and March-October 2017, while

the KIT EM27 (owned by the Karlsruhe Institute of Technology) was operated April-October 2018. Regular characterization of the Instrument Line Shape (ILS) for each EM27/SUN spectrometer was used to monitor instrument performance over time (see supplemental materials section 1). The LANL EM27 was regularly compared to the Caltech TCCON spectrometer in side-by-side observations and was used as a transfer standard to rescale EM27/SUN retrievals in Fairbanks to the TCCON trace-gas scale (see supplemental materials section 1). Tight correlations between the LANL and KIT EM27/SUN spectrometers, and

between the LANL EM27 and Caltech TCCON instrument, suggest that, given appropriate bias correction, observations from either of the two EM27/SUN spectrometers are relatively interchangeable with TCCON observations.

    Retrievals from TCCON sites are vetted with careful quality control standards before being archived publicly (Kivi et al., 2017; Wunch et al., 2017a). The GGG2014 retrieval algorithm is used to retrieve $X_{CO2}$ from TCCON and EM27/SUN observations (Wunch et al., 2015) with some input modifications introduced in EM27/SUN retrievals by EGI, discussed by Hedelius





et al. (2016) and sourced from Hedelius and Wennberg (2017). Quality controls applied to EM27/SUN retrievals follow those outlined by Hedelius et al. (2016) including an upper bound on sza at $82°$ and an upper bound on $X_{CO2}$ retrieval error at 5 ppm. In addition to the quality controls suggested by Hedelius et al. (2016), a lower bound is set on the average solar intensity (SIA) in EM27/SUN retrievals at 90 AU. After quality control filtering, EM27/SUN retrievals are smoothed by eliminating retrieved

$X_{CO2}$ that deviates by more than 1 ppm from a five-point moving average (spectra are collected approximately every 10-15 seconds). Throughout this paper, all time aggregations of retrieved $X_{CO2}$ from ground-based observations were weighted by the inverse of the retrieval error using

$$\bar{x} = \frac{\sum_i x_i x_{err,i}^{-2}}{\sum_i x_{err,i}^{-2}} \qquad (1)$$

where $x_i$ is the retrieved $X_{CO2}$ of the $i$th retrieval in the aggregation interval and $x_{err,i}$ is the corresponding retrieval error.

OCO-2 observations were retrieved with the ACOS B9 retrieval algorithm and retrievals for this study were obtained from OCO-2 B9 Lite files (OCO-2 Science Team/Michael Gunson, Annmarie Eldering, 2018), which have been initially screened for cloud cover as described by Taylor et al. (2016) and bias corrected as described by Osterman et al. (2018). Only OCO-2 soundings over land are included in this analysis (with land_fraction =100), and the standard bias correction to TCCON is applied to all OCO-2 retrievals of $X_{CO2}$ unless otherwise stated. Following the coincidence criteria defined by Wunch

et al. (2017b), OCO-2 soundings were considered coincident to ground sites if they fall within a $5°$ latitude by $10°$ longitude box centered on the ground site and occur on the same day as the corresponding ground measurements. At Fairbanks and Sodankylä OCO-2 observations consistently occur within approximately 30 minutes of local solar noon and at East Trout Lake they occur within approximately one hour of local solar noon. Therefore we define a daily ground-based reference value for $X_{CO2}$ (referred to as the near noon ground measurement or NNG), which is the error weighted average (see Eq. 1) of

ground-based $X_{CO2}$ collected within two hours of local solar noon.

## 3    Results

### 3.1    OCO-2 quality control filtering

Three different sets of quality control filters were applied to OCO-2 high latitude retrievals in this study, and are defined in Table 1 (see supplemental materials or Osterman et al. (2018) for definitions of quality control parameters). Two of these three

sets of quality control filters are recommended by the OCO Science Team for ACOS B8 retrievals (B8 QC) and ACOS B9 retrievals (B9 QC), and are summarized by the binary variables xco2_quality_flag_b8 and xco2_quality_flag in the OCO-2 B9 Lite files (Osterman et al., 2018; O'Dell et al., 2018). Methods for selecting quality thresholds and details on the B8 QC filters are discussed by O'Dell et al. (2018). Improvements in pointing accuracy in ACOS B9 (Kiel et al., 2019), as well as a careful reevaluation of quality control parameters, allowed for intentionally more permissive quality thresholds in B9 QC than those in

B8 QC, and this resulted in a substantial increase in data throughput over regions, such as the Boreal Forest and high latitudes in general, that have been sparsely represented under past OCO-2 ACOS spectral fitting and quality control regimes. The third set of quality control filters (Boreal QC) were determined by evaluating quality control histograms like those presented by





O'Dell et al. (2018) with "truth" as the NNG observations from the three Boreal Forest sites (see Appendix Fig. A1). Scatter plots of bias in $X_{CO2}$ ($\Delta X_{CO2} \equiv$ OCO-2 - NNG) against various retrieval parameters were also considered as a way to search for groupings of bias outliers that could be eliminated with small changes in quality control thresholds. The Boreal QC were set with the goal of maximizing data throughput for high latitude Boreal Forest sites in spring and autumn, while maintaining

acceptable ranges of bias at Boreal Forest sites. Additional retrieval parameters, not used in B8 or B9 QC, were also considered for the Boreal QC that relate to challenges in high latitude observations, including the difference between retrieved and a priori temperature (deltaT), solar zenith angle (sza), $X_{CO2}$ retrieval uncertainty (xco2_uncertainty), and total column water vapor (tcwv).

Changes to thresholds for albedo in the strong $CO_2$ band (albedo_sco2), the quality of the spectral fit in the weak $CO_2$

band (rms_rel_wco2), and the standard deviation in surface elevation in the satellite field of view (altitude_stddev) were major contributors to the increase in passable high latitude retrievals with B9 QC relative to B8 QC (compare Fig. 2 and Fig. 3). In the Boreal QC, ranges of acceptable values are expanded from those in the B9 QC for the ratios of single band retrievals of $CO_2$ (co2_ratio) and $H_2O$ (h2o_ratio), and the quality of the spectral fit in the weak $CO_2$ band (rms_rel_wco2). Albedo in the strong $CO_2$ band (albedo_sco2) is not used as a QC filter in the Boreal QC because it seemed that problematic data

with low albedo_sco2 could be screened by other QC filters, and there was no evidence that low albedo_sco2 was explicitly correlated to larger OCO-2 biases at Boreal Forest sites (see quality control plots in Appendix A1). In fact, increases in bias and retrieval standard deviation were more often associated with high albedo in the strong $CO_2$ band, rather than low values. More conservative thresholds were placed on the slope of albedo in the strong $CO_2$ band given by the continuum fit (albedo_slope_sco2) than were previously used in the B8 QC or the B9 QC due to observed increases in the standard deviation

of retrievals and larger negative biases specifically associated with more negative albedo slopes. One possible explanation for this observation is that certain surface types that are more prevalent in the Boreal Forest are not correctly modeled by the ACOS B9 algorithm, and this could be related to snow covered surfaces. We expect that introducing a polynomial fit to the albedo in each band, rather than a linear fit, could improve the accuracy of modeled surface albedo in future ACOS versions and potentially result in reduced high latitude biases. Thresholds for the difference between retrieved surface pressure and

a priori surface pressure at the pointing location of the $O_2A$ band (dp_o2a) remained the same in Boreal QC as in B9 QC, while thresholds for the difference between retrieved surface pressure and a priori surface pressure at the pointing location of the strong $CO_2$ band (dp_sco2) were made marginally more conservative. Kiel et al. (2019) discuss the pointing errors and other long term challenges with surface pressure bias in OCO-2 retrievals that lead to the addition of the dp_o2a and dp_sco2 parameters, in which there is one retrieved surface pressure and a separate a priori surface pressure defined for each band. The

aerosol optical depth (aod) parameters are mostly the same in the Boreal QC as in the B9 QC with the exceptions that total aod (aod_total) and the combined dust, water, and seasalt aod (dws) were removed in the Boreal QC because these seemed superfluous after applying other aod filters. While the range of acceptable values for the difference between retrieved and a priori vertical $CO_2$ gradient (co2_grad_del) is nearly the same in the B9 QC as in the Boreal QC, the range of values is shifted up. This choice was made based on the distribution for co2_grad_del for the Boreal Forest sites, and the difference may be

attributed to the use of a regional dataset for Boreal QC rather than a global dataset for B9 QC. As previously mentioned, several





**Table 1.** Quality control thresholds currently used to determine the B8 QC (xco2_quality_flag_b8=0 in OCO-2 Lite files), B9 QC (xco2_quality_flag=0 in OCO-2 Lite files), and Boreal QC (new proposed thresholds for terrestrial high latitude sites). Descriptions of QC parameters in this table can be found in supplemental materials (Table S2) or from Osterman et al. (2018).

| Name | B8 QC | | B9 QC | | Boreal QC |
|------|-------|---|-------|---|-----------|
| | glint and nadir | target | glint and nadir | target | all modes |
| co2_ratio | [1.00, 1.025] | | [1.00, 1.023] | | [1.00, 1.028] |
| h2o_ratio | [0.88, 1.01] | | [0.88, 1.01] | | [0.80, 1.02] |
| altitude_stddev | [0, 60] | [0, 20] | [0, 110] | | [0, 110] |
| max_declocking_wco2 | [0.0, 0.75] | | – | | – |
| dp | [-6, 14] | | – | | – |
| dp_sco2 | – | | [-10, 12] | | [-9, 12] |
| dp_o2a | – | | [-8, 11] | | [-8, 11] |
| dp_abp | [-10, 13] | [-10, 50] | [-12, 16] | [-12, 50] | [-12, 20] |
| co2_grad_del | [-80, 100] | | [-60, 85] | | [-50, 100] |
| albedo_sco2 | [0.05, 0.60] | | [0.03, 0.60] | | – |
| rms_rel_wco2 | [0.0, 0.22] | | [0.0, 0.28] | | [0.0, 0.35] |
| rms_rel_sco2 | – | | [0.0, 0.45] | | – |
| s31 | [0.03, 0.4] | | – | | – |
| albedo_slope_sco2 | [-0.00018, 0.001] | | [-0.00013, 0.001] | | [-0.0001, 0.0004] |
| aod_total | [0.0, 0.5] | | [0.0, 0.5] | | – |
| dws | [0.0, 0.25] | | [0.0, 0.25] | | – |
| aod_water | [0.0005, 0.1] | | [0.0005, 0.1] | | [0.0005, 0.1] |
| aod_ice | [0.0, 0.04] | | [0.0, 0.04] | | [0.0, 0.04] |
| ice_height | [-0.5, 0.45] | | [-0.5, 0.5] | | [-0.5, 0.5] |
| aod_sulfate + aod_oc | [0.0, 0.3] | | – | | – |
| aod_strataer | [0.0, 0.02] | | [0.0002, 0.02] | | [0.0002, 0.02] |
| aod_oc | [0.0, 0.08] | | [0.0, 0.20] | | [0.0, 0.20] |
| aod_seasalt | [0.0, 0.125] | | [0.0, 0.125] | | [0.0, 0.125] |
| deltaT | – | | – | | [-1, 1] |
| sza | – | | – | | [0, 70] |
| xco2_uncertainty | – | | – | | [0, 1.5] |
| tcwv | – | | – | | [3, 40] |

parameters were used to define quality control filters in the Boreal QC that were not included in the parameters for B8 QC and B9 QC. A threshold for sza was introduced in the Boreal QC, and was chosen to restrict data furthest North to the months of March through November. Potential challenges with data at high sza are discussed in the introduction of this paper and high





sza was found to be correlated with larger negative OCO-2 biases at Boreal Forest sites. The sza threshold in the Boreal QC only screens approximately 0.5% of retrievals that manage to get through the other Boreal QC filters. The thresholds placed on the difference between retrieved and a priori temperature (deltaT) and total column water vapor (tcwv) were chosen because very low atmospheric water vapor or large differences between retrieved and modeled temperatures are likely to correspond

with cold weather and snow cover. In particular, before the application of quality control filters, large negative biases in OCO-2 retrievals were found to be associated with low values of tcwv (discussed in more detail in section 4.5). Although the majority of biased retrievals with low tcwv are screened out by other quality filters, this filter helped to remove a small number of outliers that pass the other QC filters (see Fig. 24). Finally, the uncertainty in retrieved $X_{CO2}$ (xco2_uncertainty) was included arbitrarily in the analysis and found to be effective in eliminating a small number of outliers that made up less than 0.05% of

retrievals not screened by other filters.

### 3.2    Effect of QC on data throughput north of 50°N

For each set of QC filters all retrievals over land north of 50°N latitude in OCO-2 Lite files were evaluated to determine how many failed to meet the quality thresholds for each parameter in each month (see Fig. 2, Fig. 3, and Fig. 4). Figures 2 and 3 show there is a clear seasonality to triggered quality filters with the majority occurring in spring and early summer.

This seasonality is slightly diminished with the B9 QC (Fig. 3) relative to the more conservative B8 QC (Fig. 2) and is only marginally manifested in the Boreal QC (Fig. 4). The reduction in the number of filtered soundings in spring with Boreal QC is largely attributable to less conservative bounds on the spectral fit quality in the weak and strong $CO_2$ bands (rms_rel_wco2, rms_rel_sco2) and the ratios of single band retrievals of $CO_2$ (co2_ratio) and $H_2O$ (h2o_ratio). In all three sets of quality control filters the parameters whose thresholds are most often triggered, resulting in the removal of data points, are the spectral

fit quality (rms_rel_wco2, rms_rel_sco2), the ratios of single band retrievals of $CO_2$ (co2_ratio) and $H_2O$ (h2o_ratio), and differences between the retrieved and various a priori surface pressures (dp_sco2, dp_o2a, dp). The fact that these parameters account for a greater abundance of flagged retrievals in spring and autumn suggests that there could be seasonal effects related to these retrieval parameters that need to be accounted for in high latitude measurements. In particular, there has been speculation that spring snow cover would result in low surface albedo in the 1.61 $\mu$m and 2.06 $\mu$m bands and patchy snow cover or snow-

free vegetation protruding from snow pack could cause variability in albedo within the satellite's field of view (Wiscombe and Warren, 1980; Boesch et al., 2011). However, after matching MODIS snow cover data to coincident OCO-2 retrievals at our Boreal Forest sites there was no clear connection found between snow cover and increased magnitudes of OCO-2 bias with or without QC filtering. It may still be the case that incongruous spatial resolution between MODIS and OCO-2 is masking the effects of snow cover on bias or that OCO-2 is only biased by snow in combination with certain other effects of cold weather

conditions that are more frequently occurring in spring.

Additional data gained from applying Boreal QC rather than B9 QC can be visualized as an increase in spatial coverage of terrestrial high latitude regions. Figures 5 and 6 show the latitude and longitude with 1° precision that are represented with passable OCO-2 retrievals in Boreal QC and are not in B9 QC. These maps point to substantial increases in spatial coverage of the Boreal Forest ($\sim$ 50°N-70°N latitude band) in the spring and autumn months with Boreal QC. This improvement in





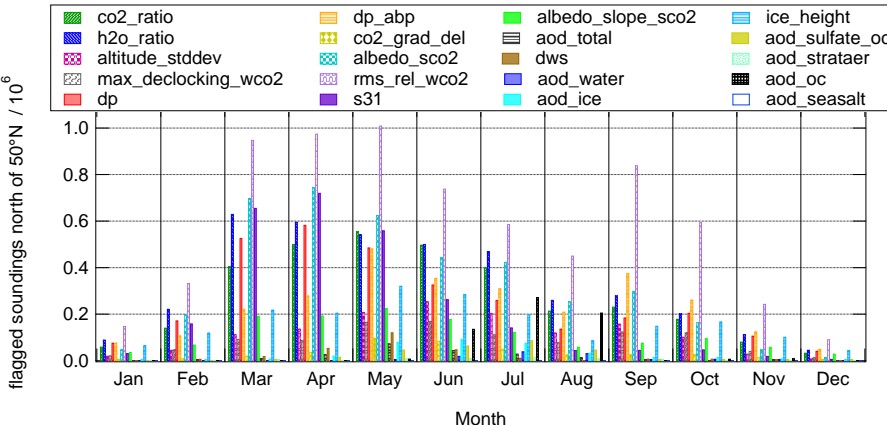

**Figure 2.** Total number of land soundings north of 50°N flagged by B8 quality filters in each month.

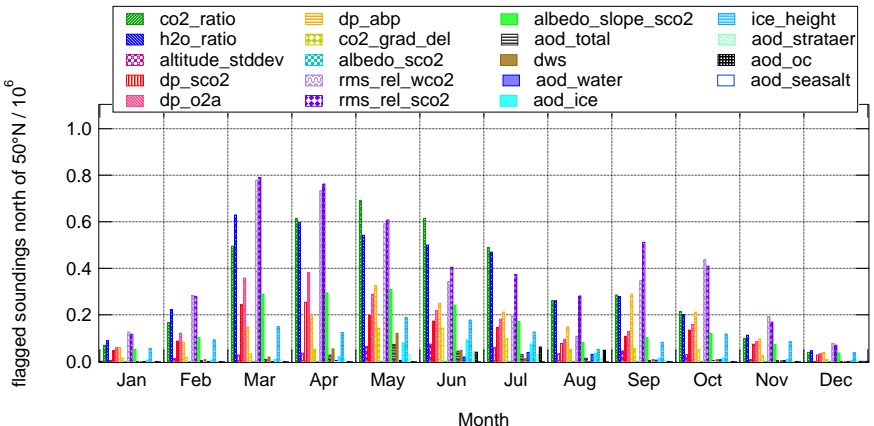

**Figure 3.** Total number of land soundings north of 50°N flagged by B9 quality filters in each month.

coverage is an important advantage of the Boreal QC for selecting OCO-2 retrievals with the goal of evaluating longitudinal trends in seasonal cycles for the Boreal Forest.

The B9 QC filters succeed in tripling the number of passed retrievals over land at high latitudes, relative to the B8 QC, and the Boreal QC allow nearly double the number of retrievals allowed by B9 QC (see Fig. 7 and the right column of Fig.

5   9). An important result of the Boreal QC is the increase in passed retrievals in May, August, and September relative to the B9 QC. While the more relaxed B9 QC allow more high latitude retrievals than B8 QC, the relative number of soundings passed from month to month remains roughly unchanged. By plotting monthly snow extent in the Northern Hemisphere, as reported by NOAA (Robinson et al., 2012), alongside monthly average sza and monthly passed soundings north of 50°N, Fig. 7 provides further evidence that some combination of sza and snow cover could be playing a role in high latitude data removal.

10   If solar zenith angles (sza) were the primary driver of seasonality in high latitude data throughput, one would expect to obtain





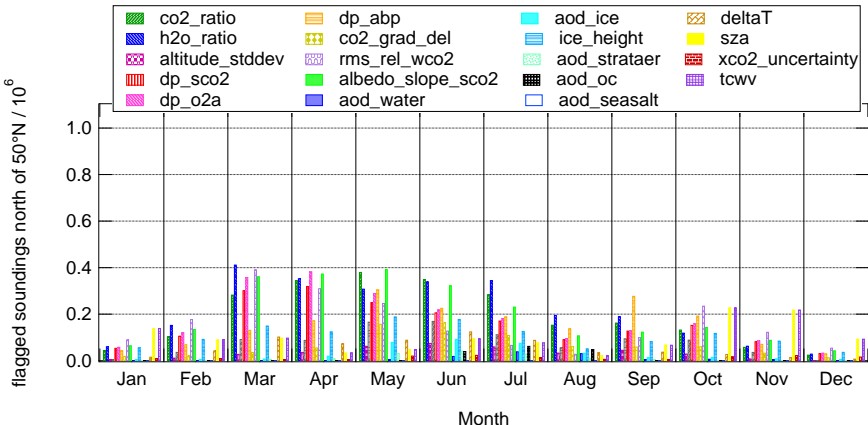

**Figure 4.** Total number of land soundings north of 50°N flagged by Boreal quality filters in each month.

approximately the same quantity of passed retrievals in May as in July, but Fig. 7 indicates that nearly twice as many high latitude retrievals pass QC filters in July. As mentioned previously in this section, additional analysis did not lead us to the conclusion that snow is the culprit in itself, but some effects from snow or differences between fresh and melting snow cannot be entirely excluded either. It remains unclear how combinations of radiative transfer effects may be contributing to increased
data removal at high latitudes in spring.

### 3.3   Comparing OCO-2 and ground-based observations

#### 3.3.1   Averaging kernel corrections

The retrieval averaging kernel represents the sensitivity of retrieved $X_{CO2}$ to enhancements at different altitudes in the atmospheric column. When comparing retrievals of $X_{CO2}$ from two different spectrometers, averaging kernels can be used to
mathematically correct for systematic sources of disagreement that result from instrumental differences. In this paper, averaging kernel corrections were applied to simulate the OCO-2 retrieval that would result by assuming the ground-based retrieval to be truth and scaling by the OCO-2 averaging kernel with an averaging kernel correction factor, $d$NNG (see supplemental materials section 2). The result of averaging kernel corrections is a set of modified ground-based measurements ($\tilde{X}_{\mathrm{NNG}}$) that are the sum of the NNG $X_{CO2}$ aggregate ($X_{\mathrm{NNG}}$) and a $d$NNG value that is uniquely calculated for each coincident OCO-2
retrieval, such that

$$\tilde{X}_{\mathrm{NNG}} = X_{\mathrm{NNG}} + d\mathrm{NNG}. \tag{2}$$

and

$$d\mathrm{NNG} = (1 - \gamma)\mathbf{h}^T \mathbf{A}_0 \mathbf{x}_a \tag{3}$$

where $\gamma$ is the scaling ratio of retrieved to a priori near noon ground-based $X_{CO2}$, $\mathbf{h}$ is the pressure weighting function, $\mathbf{A}_0$ is
the OCO-2 averaging kernel, and $\mathbf{x}_a$ is the a priori $CO_2$ profile (see supplemental materials section 2 for details). Averaging



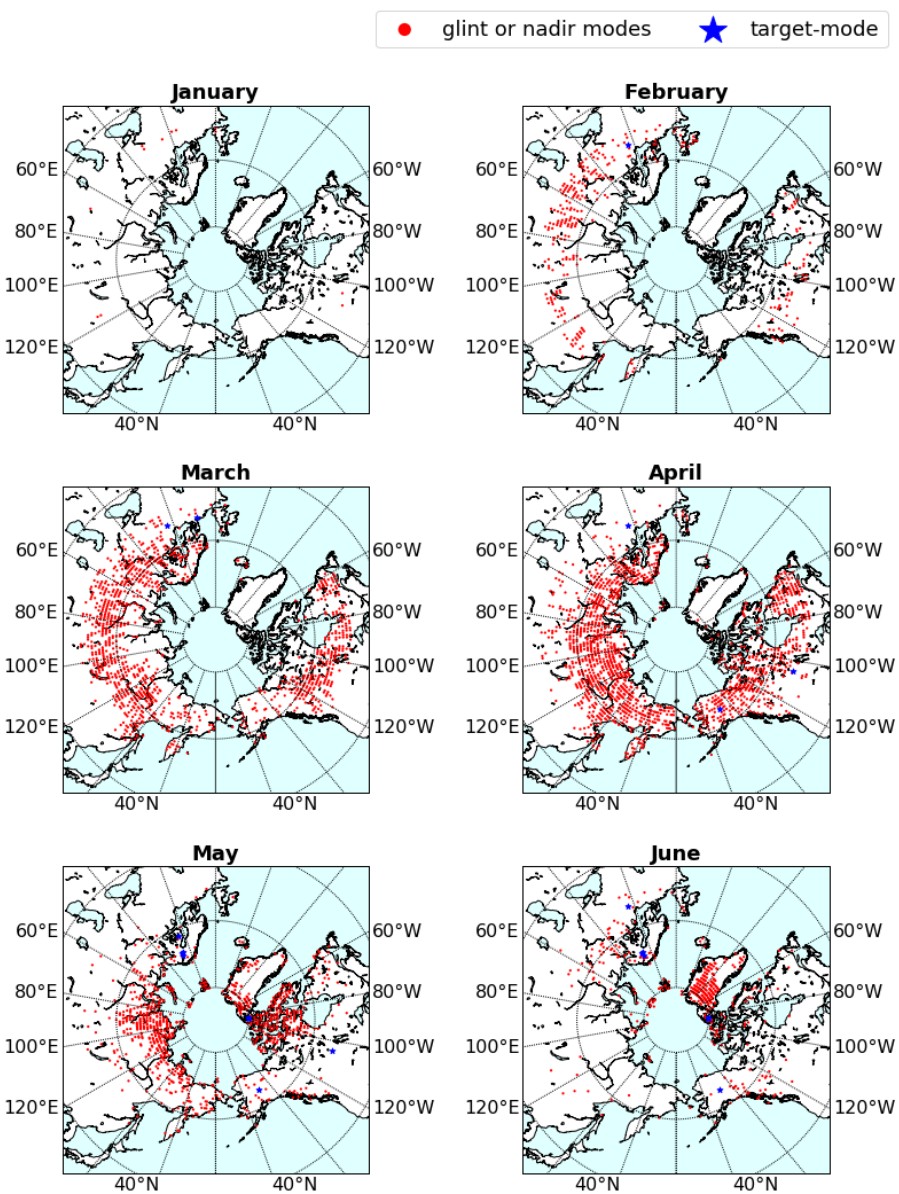

**Figure 5.** Maps of additional data coverage gained in each month (January to June, 2014 to 2018) from applying Boreal QC instead of B9 QC, with 1° resolution.

kernel correction factors ($d$NNG) display some seasonal variability, the ratio of retrieved to a priori ground-based $X_{CO2}$ was found to be the dominant term causing this seasonality (see supplemental materials and compare $d$NNG in the top row of Fig. S2 to $(1 − \gamma)$ in the third row of Fig. S2). Any seasonality introduced by averaging kernel corrections appears to be on too



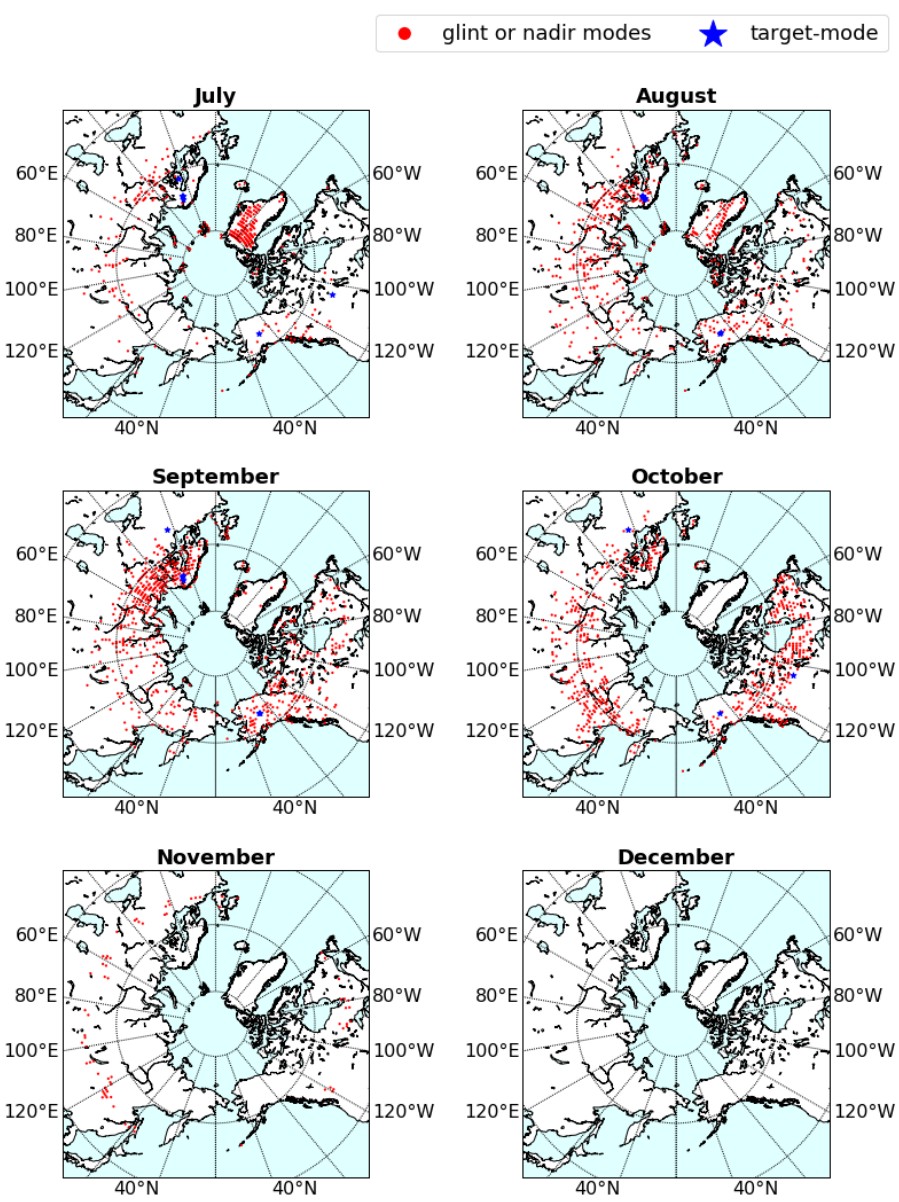

**Figure 6.** Maps of additional data coverage gained in each month (July to December, 2014 to 2018) from applying Boreal QC instead of B9 QC, with 1° resolution.

small a scale to explain seasonal variability observed in the following sections. Note that in the remainder of this paper East Trout Lake, Sodankylä, and Fairbanks are abbreviated in figures to ETL, SOD, and UAF, respectively.





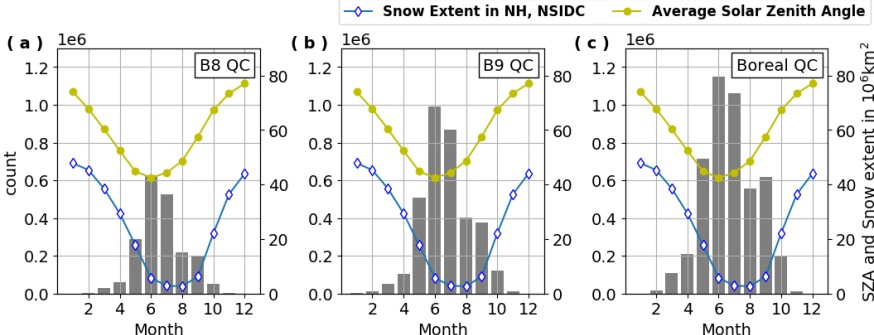

**Figure 7.** Total number of land soundings north of $50°$N that pass QC filters in each month, plotted along with the average sza reported in OCO-2 retrievals that pass in each month and the monthly snow extent in the Northern Hemisphere averaged over 2014-2018 as reported in the NOAA database (Robinson et al., 2012).

### 3.3.2   Biases by site, viewing mode, and QC method

To evaluate the effects of these three quality control methods on observed biases and data throughput, NNG observations with averaging kernel corrections applied ($\tilde{X}_{\mathrm{NNG}}$) were compared against coincident OCO-2 retrievals at three sites in the Boreal Forest (East Trout Lake, Fairbanks, and Sodankylä) (see section 2 for definitions of NNG and coincidence criteria). Daily

averages for the complete set of coincident OCO-2 retrievals obtained with Boreal QC and corresponding $\tilde{X}_{\mathrm{NNG}}$ are shown in Fig. 8. While the OCO-2 and NNG observations in Fig. 8 appear to be in close agreement on most days there are a few outliers in the OCO-2 daily averages in spring and autumn that may contribute to a potential seasonal dependence in bias. Figure 9 provides an overview of the full datasets for each site including total average bias, standard deviation in bias, and data throughput, sorted by satellite observing mode and quality control method. Note that bias is defined as $\Delta X_{CO2} \equiv$ (OCO-2

retrieval)$_i - (\tilde{X}_{\mathrm{NNG}})_i$ for each coincident OCO-2 sounding, so that a negative bias indicates that OCO-2 retrievals are lower than NNG and a positive bias indicates that OCO-2 retrievals are higher than NNG. At all three sites, target mode retrievals had 0.1 ppm to 0.5 ppm lower standard deviation than glint or nadir retrievals, which may indicate the introduction of proximity bias (i.e., soundings further from the ground site contributing larger bias). If proximity bias is an important source of bias, one may expect that target mode retrievals would also have lower average biases than glint and nadir retrievals. Results from

East Trout Lake (ETL) and Fairbanks (UAF) meet this expectation, with the exception of results from the B8 QC at East Trout Lake. Only at Sodankylä the average biases in target mode retrievals substantially exceed the average biases in glint or nadir retrievals, warranting further investigation of target observations at Sodankylä. In particular, the B8 QC results in an average bias in target mode soundings at Sodankylä that is at least twice that observed at East Trout Lake or Fairbanks, and because this increase is accompanied by increased standard deviation in target mode biases at Sodankylä, it could indicate influence from

outliers. While the allowance of additional data switching from B8 QC to B9 QC or from B9 QC to Boreal QC tends to be accompanied by an approximate increase in the standard deviation in biases of 0.1 ppm to 0.3 ppm, average biases at the three

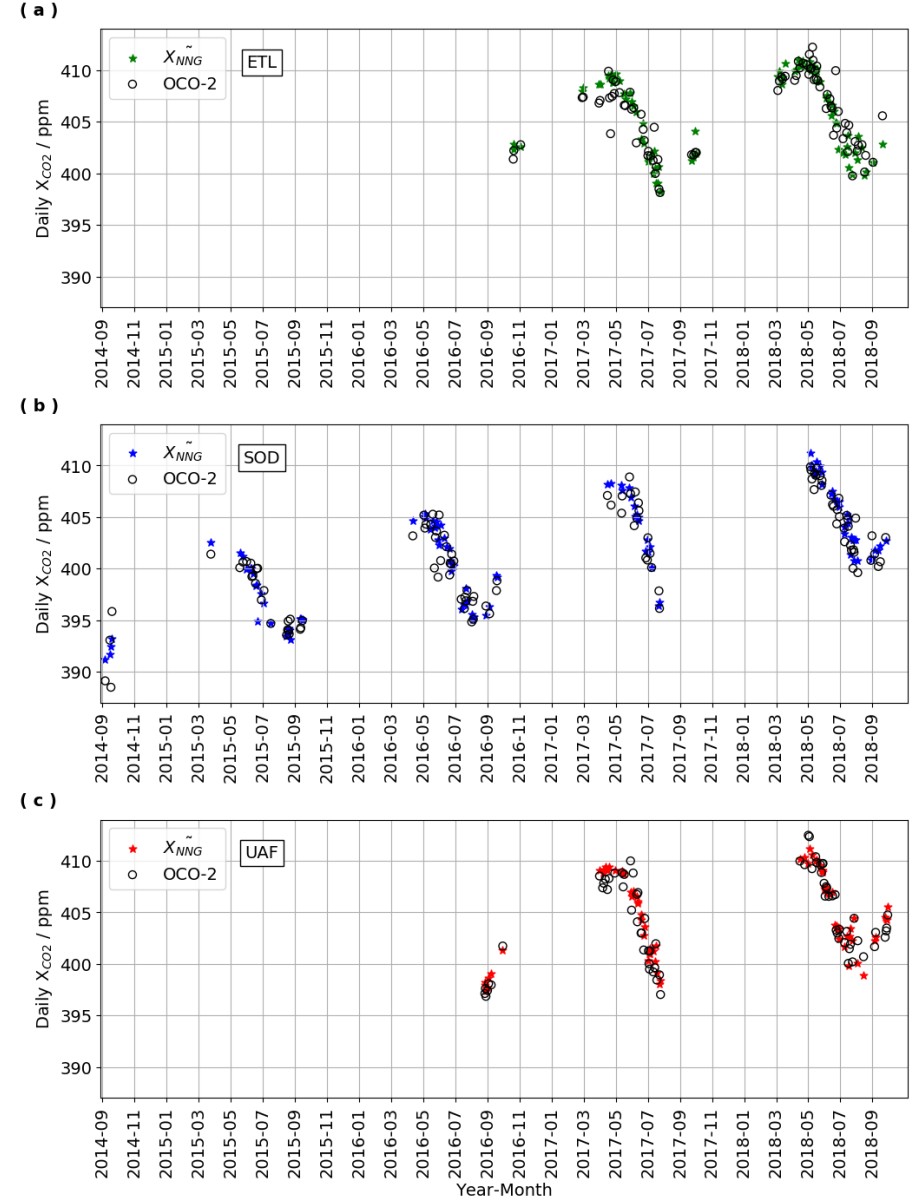

**Figure 8.** Time-series of ground-based and satellite-based data at each Boreal Forest site. Consisting of daily averages of OCO-2 coincident soundings filtered with Boreal QC, alongside corresponding daily averages of NNG with averaging kernel corrections to OCO-2 applied as described in section 3.3.1 ($\tilde{X}_{\mathrm{NNG}}$).

sites are not consistently larger with the Boreal QC (see left and center columns of Fig. 9). In considering the use of Boreal QC for certain science applications at high latitudes, the introduction of additional scatter should be weighed against the large increase in usable retrievals as shown in the right column (panels (c), (f), and (i)) of Fig. 9 and in Fig. 7.

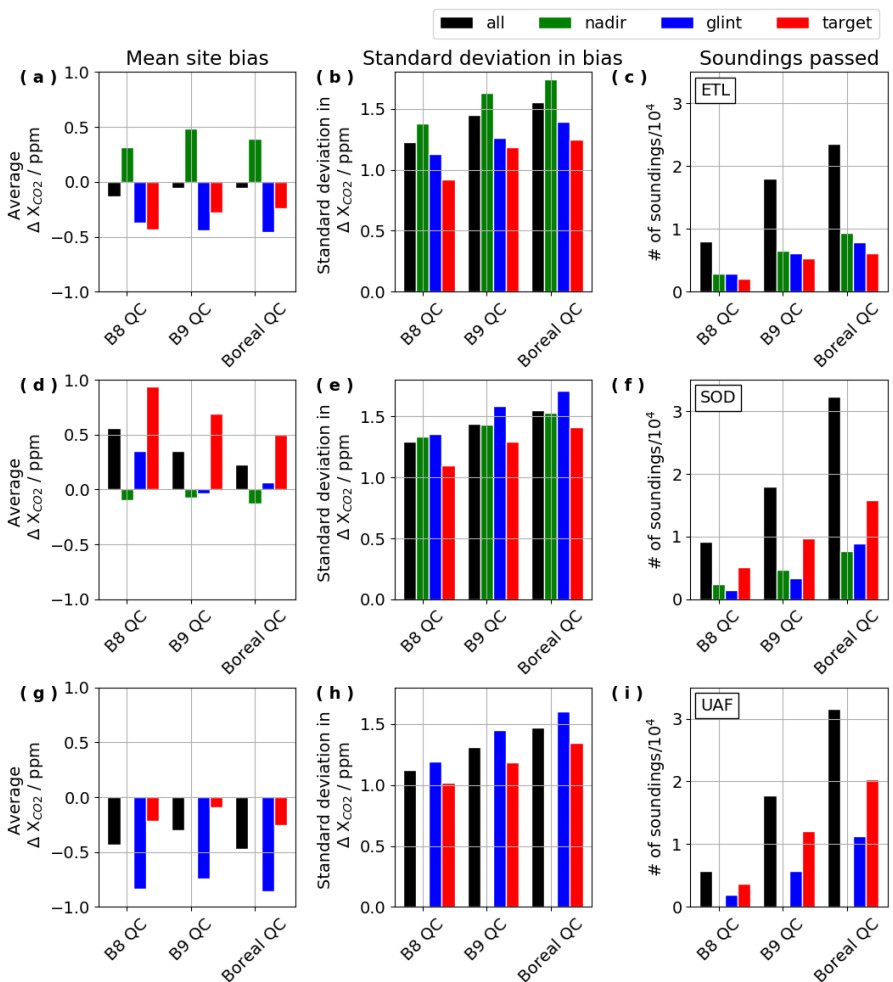

**Figure 9.** Average bias, standard deviation in bias, and number of passed soundings sorted by viewing geometry and quality control method and considering all coincident soundings at each of the three Boreal forest sites. Note that there are no coincident nadir soundings for Fairbanks due to the satellites operational design, which favors glint observations in orbits primarily over oceans. Note that bias is defined as $(\Delta X_{CO2} \equiv (\text{OCO-2 retrieval})_i - (\tilde{X}_{\text{NNG}})_i)$ for each coincident OCO-2 sounding.

## 3.4 Seasonal variability in bias

High latitude regions experience a higher degree of seasonality in many climate and environment variables than mid-latitude regions, and one of our primary motivations in this study is expanding our ability to evaluate $CO_2$ seasonality in the Boreal Forest. Considering the total average and standard deviation in biases for all coincident soundings, as in Fig. 9, can obscure seasonal variability in biases that contribute to uncertainty in characterizing seasonal cycles of $CO_2$ obtained from satellite observations. Figure 10 shows monthly average biases and monthly standard deviation in biases, considered for each site and





each set of QC filters. Under all three sets of QC filters we observe seasonal trends in biases, which are more pronounced at East Trout Lake and Sodankylä than at Fairbanks. The observed seasonal variability is characterized by more positive biases in mid to late summer and more negative biases in spring and autumn, which may cause satellite-based estimates of seasonal amplitude and timing to differ from ground-based estimates of these seasonal parameters. For most months the Boreal QC do
not result in substantial increases in the absolute value of average monthly biases relative to B9 QC, but in April the Boreal QC yields larger negative biases than B9 QC at Fairbanks and at Sodankylä large negative biases are observed in the Boreal QC without any counterpart in the B8 or B9 QC to compare to. In Fairbanks the average April bias drops from -0.59 ppm, with the B9 QC, to -1.15 ppm, with the Boreal QC. Figure 18, panel (a), shows that the difference in the absolute values of the average April bias in Fairbanks is the largest difference in absolute monthly biases when comparing the Boreal and B9 QC methods. At
Sodankylä April data are only allowed by the Boreal QC, while B8 QC and B9 QC filter out all coincident OCO-2 retrievals, but the negative average bias obtained with Boreal QC at Sodankylä in April represents the maximum absolute monthly bias of any month, site, or QC method. Figure 10 also shows larger negative biases with the Boreal QC at Fairbanks in August, September, and October in which the Boreal QC yields average monthly biases descending from -0.46 ppm to -0.72 ppm in consecutive months, while the B9 QC yields average biases descending from -0.08 ppm to -0.43 ppm. At East Trout Lake and
Sodankylä, the monthly biases are only marginally different between the Boreal QC and B9 QC, and the Boreal QC resulted in slightly smaller monthly biases in March, June, and November at East Trout Lake and in August at Sodankylä (see Fig.10 and Fig. 18 panel (a)).

    Monthly bias distributions are visualized with box-plots for each site and set of QC filters in Fig. 11 to further elucidate potential seasonal trends in OCO-2 biases. Figure 11, panels (a), (b), and (c) show that East Trout Lake has the most pronounced
seasonal variability in biases and the trends observed are similar for all three sets of QC filters. Figure 11 panel (f) suggests there is a slight seasonal trend at Sodankylä with the Boreal QC that appears when March and April soundings are included. Overall, the monthly bias distributions also serve to emphasize the similarity in results from the different QC methods.

## 3.5 A modified OCO-2 bias correction with T700

The seasonal dependence of $\Delta X_{CO2}$ described in the previous section was found to be largely induced by the OCO-2 bias
correction, and is not apparent in $\Delta X_{CO2}$ calculated with un-bias-corrected OCO-2 retrievals (see Fig. 12). In OCO-2 B9 retrievals, the B9 bias correction (B9 bc) for soundings over land is defined by Osterman et al. (2018) as

$$X_{CO2,corrected} = \frac{X_{CO2,raw} - foot + 0.9(\text{dpfrac}) + 9.0(\text{dws}) + 0.029(\text{co2\_grad\_del} - 15)}{0.9954} \qquad (4)$$

with a footprint bias correction term, $foot$, an overall divisor to agree with TCCON, and parameter dependent terms adjusting based on a modified parameterization of the retrieved surface pressure bias defined by Kiel et al. (2019) (dpfrac), the sum
of dust, water, and seasalt aods (dws), and the difference between retrieved and a priori vertical gradients in the $CO_2$ profile (co2\_grad\_del). Of the terms in the B9 bc, dpfrac was the only one found to have seasonal variability at Boreal Forest sites that was similar to that observed in $\Delta X_{CO2}$ with the OCO-2 bias correction (see Fig. 11 and 28). As will be discussed in section 4.6, all versions of the residual in retrieved surface pressure relative to a priori surface pressure (dpfrac, dp, dp\_o2a, dp\_sco2)



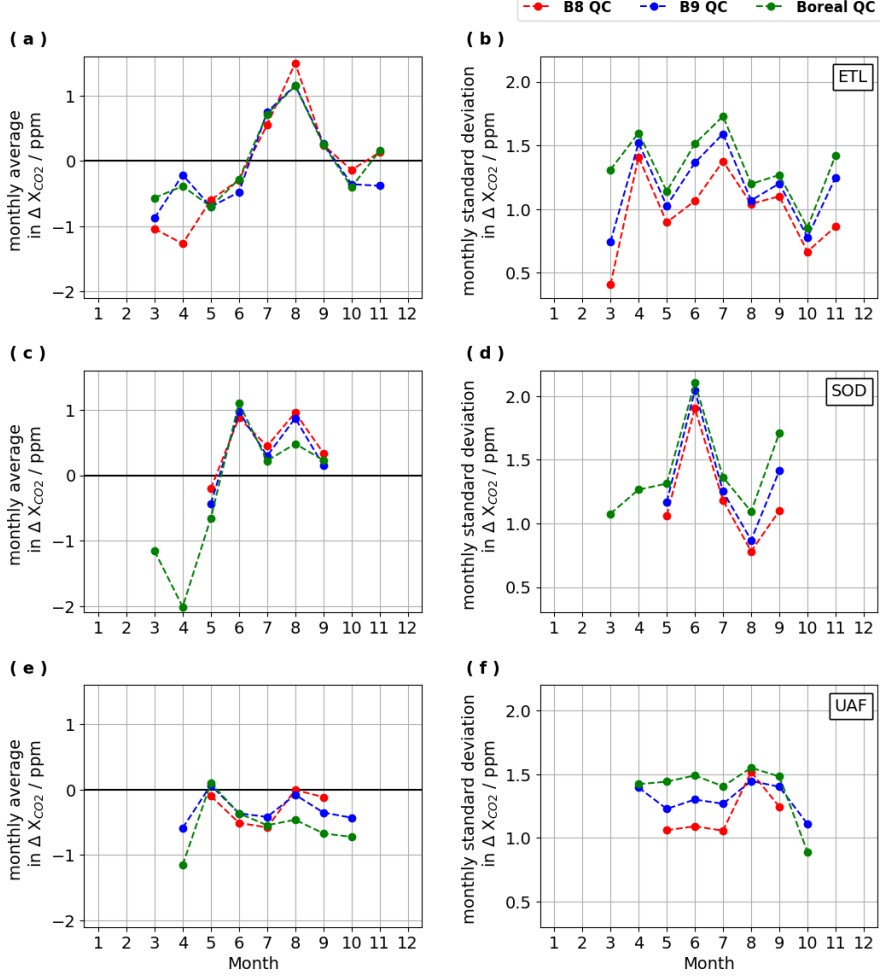

**Figure 10.** Monthly average bias (left) and standard deviation in biases (right) of coincident OCO-2 soundings at each of the three Boreal Forest sites and with each of the QC methods. Note that bias is defined as ($\Delta X_{CO2} \equiv$ (OCO-2 retrieval)$_i$ − ($\tilde{X}_{\mathrm{NNG}})_i$) for each coincident OCO-2 sounding.

have seasonal variability that can be at least partially attributed to temperature dependence, so we propose new OCO-2 bias corrections with a term for temperature at 700 hPa (T700) to correct for the temperature dependence in dpfrac and dp. To calculate the temperature dependent modification to the B9 bc we consider the linear regressions for dpfrac as a function of T700 in each of the satellite viewing modes for soundings over land north of 50°N that pass Boreal QC (see Fig. 13). Then the

5 regression coefficients for the different viewing modes are combined into average slope and average y-intercept with weighting



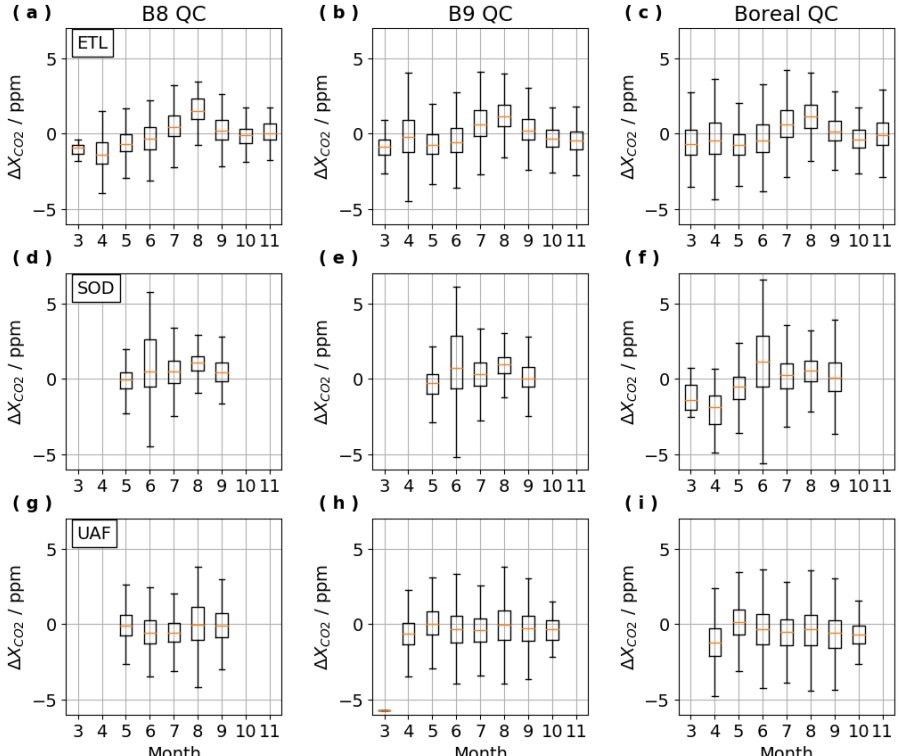

**Figure 11.** Box-plots of distributions of monthly biases for each Boreal Forest site and for each QC. Box-plots show the median in the center of the box, the first and third quartile as the bottom and top of the box and the full range of data values as the bars extending above and below the box. Note that bias is defined as ($\Delta X_{CO2} \equiv$ (OCO-2 retrieval)$_i - (\tilde{X}_{\mathrm{NNG}})_i$) for each coincident OCO-2 sounding.

by the fractional abundance of retrievals in that mode to obtain an alternative B9 bias correction (B9 abc), as follows:

$$X_{CO2,corrected} = \frac{X_{CO2,raw} - foot + 0.9(\mathrm{dpfrac} - (0.068(\mathrm{T700}) - 19.03)) + 9.0(\mathrm{dws}) + 0.029(\mathrm{co2\_grad\_del} - 15)}{0.9954} \quad (5)$$

$$= \frac{X_{CO2,raw} - foot + 0.9(\mathrm{dpfrac}) - 0.0612(\mathrm{T700} - 279.9) + 9.0(\mathrm{dws}) + 0.029(\mathrm{co2\_grad\_del} - 15)}{0.9954}. \quad (6)$$

An alternative B8 bias correction (B8 abc) was also constructed using linear regression terms for the difference between the

5    retrieved and a priori surface pressure from GEOS5-FP-IT (dp) as a function of T700 in Fig. 14, as follows:

$$X_{CO2,corrected} = \frac{X_{CO2,raw} - foot + 0.36(\mathrm{dp} - (0.165(\mathrm{T700}) - 45.84)) + 8.5(\mathrm{dws}) + 0.029(\mathrm{co2\_grad\_del} - 15)}{0.9958} \quad (7)$$

$$= \frac{X_{CO2,raw} - foot + 0.36(\mathrm{dp}) - 0.0594(\mathrm{T700} - 277.8) + 8.5(\mathrm{dws}) + 0.029(\mathrm{co2\_grad\_del} - 15)}{0.9958}. \quad (8)$$

Applying a modification of the B8 bias correction is consistent with the fact that spectroscopy and most aspects of the radiative transfer model remained the same when the ACOS version was updated from B8 to B9. Ideally, a global bias correction would

10    be constructed to include temperature as a component of a broader analysis that considers contributions to the OCO-2 bias in





a more holistic context. In this way, the bias correction would be more uniform and widely applicable, while the effects of potential parameter covariance or other influences that we are unable to control for in this regional analysis can be mitigated. That being said, the results that follow suggest that correcting for temperature dependence could be effective in reducing seasonality in OCO-2 bias over the Boreal Forest.

The second column of Fig. 15 (panels (b), (f), and (j)), in addition to results in Fig. 16 and Fig. 17, show that seasonally dependent variability in biases is reduced when the dpfrac term is removed from the B9 bc, but both monthly and overall standard deviations in biases are increased. Without the dpfrac term in the B9 bc, monthly biases in March and April at Sodankylä and in April at Fairbanks are substantially reduced, and month to month variability at East Trout Lake is also reduced. Replacing the dpfrac or dp term with a T700 modification, as in B9 abc and B8 abc (Eq. 6 and Eq. 8), results in

lower monthly standard deviations in biases than those obtained in the B9 bc with the dpfrac term removed, and that are nearly equivalent to those obtained with the standard B9 bc (Eq. 4). While some of the seasonal shape is reintroduced with the B9 abc and the B8 abc, biases are still reduced in spring and fall relative to the B9 bc (see Fig. 16). The combined results of Fig. 16 with the total average biases and total standard deviations in biases shown in Fig. 17 suggest that the B8 abc is slightly more effective than B9 abc in reducing seasonal variability in bias, reducing total average bias, and reducing total standard

deviations in biases. Figure 17 demonstrates that for all sites and viewing modes, most of the total average biases with the B8 abc are within $\pm 0.5$ ppm. In particular, the B8 abc results in reduced average bias in target mode soundings at all three sites, as well as in nadir soundings at Sodankylä and in glint soundings at East Trout Lake and Fairbanks. The B8 abc did result in slight increases in total average biases in nadir soundings at East Trout Lake and in glint soundings at Sodankylä (see Fig. 17). However, with the B9 abc, average biases in all modes at Sodankylä, and in nadir and target retrievals at East Trout Lake, are

nearly doubled relative to the standard B9 bc.

## 4   Discussion

Results from this analysis have revealed that modified QC filters for OCO-2 retrievals have the potential to recover large quantities of previously screened terrestrial high latitude observations and provide double or triple the number of retrievals for use in scientific studies of high latitude regions. This increase in data throughput is accompanied by only minor changes in

average bias and increases in standard deviations in bias of approximately 0.3 ppm or less (see Fig. 9 and Fig. 18). Through monthly comparisons between coincident OCO-2 retrievals and ground-based measurements at three Boreal Forest sites, biases ($\Delta X_{CO2}$) were found to exhibit some seasonal variability that is mostly independent of the QC method applied (see Fig. 10 and Fig. 11). Seasonally dependent biases can be challenging to correct and can ultimately result in biases between satellite-based and ground-based estimates of seasonal cycle parameters such as amplitude and timing. Therefore, it is essential that

any seasonality in biases be characterized and potential sources be identified. Both the B9 abc in Eq. 6 and the B8 abc in Eq. 8 result in reduced monthly average biases in spring and fall months, and the standard deviations in biases are nearly equivalent to those obtained with the standard B9 bc in Eq. 4. The B8 abc was found to be slightly more effective than the B9 abc in reducing the seasonal curvature in monthly biases at East Trout Lake and Sodankylä, which could allow for lower





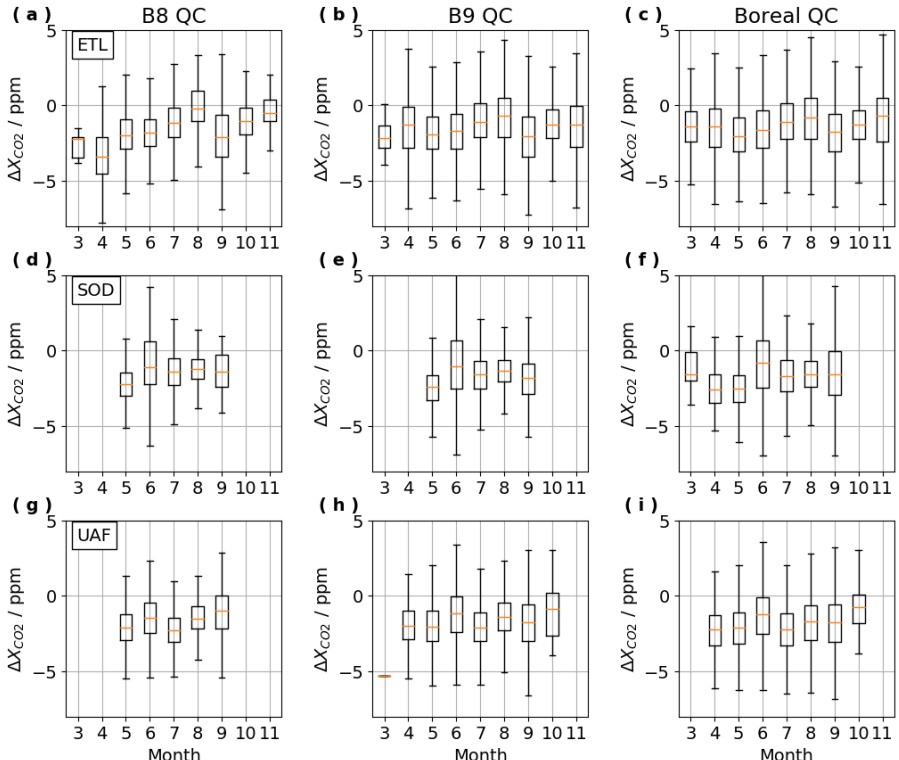

**Figure 12.** Box-plots of distributions of monthly biases for each Boreal Forest site and for each QC, without the OCO-2 bias correction. Note shift in y-axis scaling relative to Fig. 11.

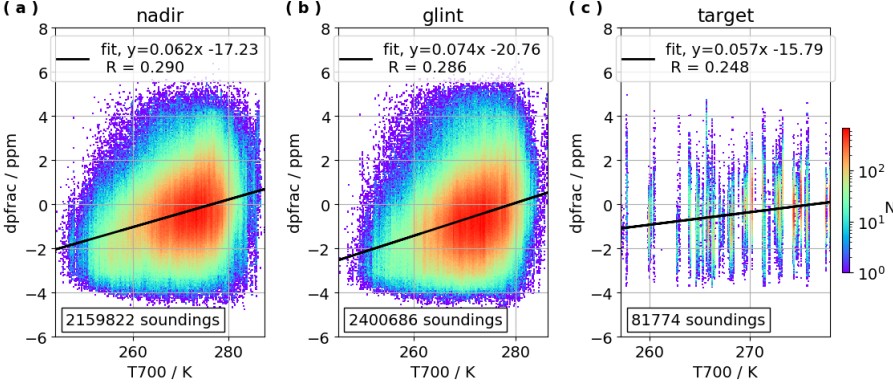

**Figure 13.** Correlations and linear regressions for dpfrac (defined in supplemental materials or by Kiel et al. (2019)) as a function of temperature at 700 hPa (T700) for all retrievals over land north of 50°N that pass Boreal QC, separated by viewing geometry.



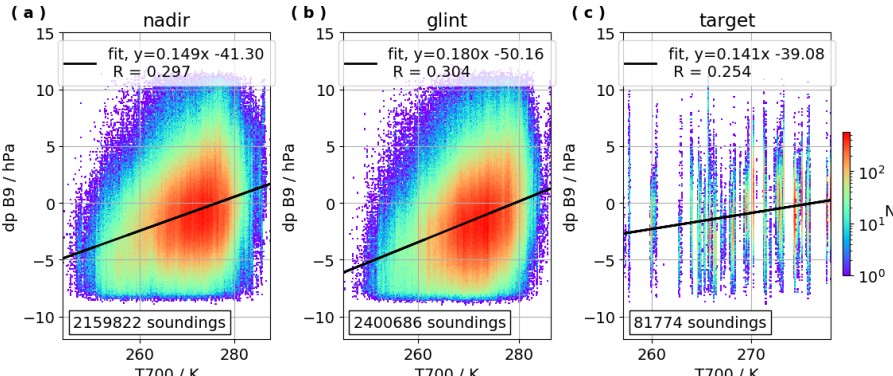

**Figure 14.** Correlations and linear regressions for the difference between retrieved and a priori surface pressure from GEOS5-FP-IT (dp) as a function of temperature at 700 hPa (T700) for all retrievals over land north of $50°$N that pass Boreal QC, separated by viewing geometry.

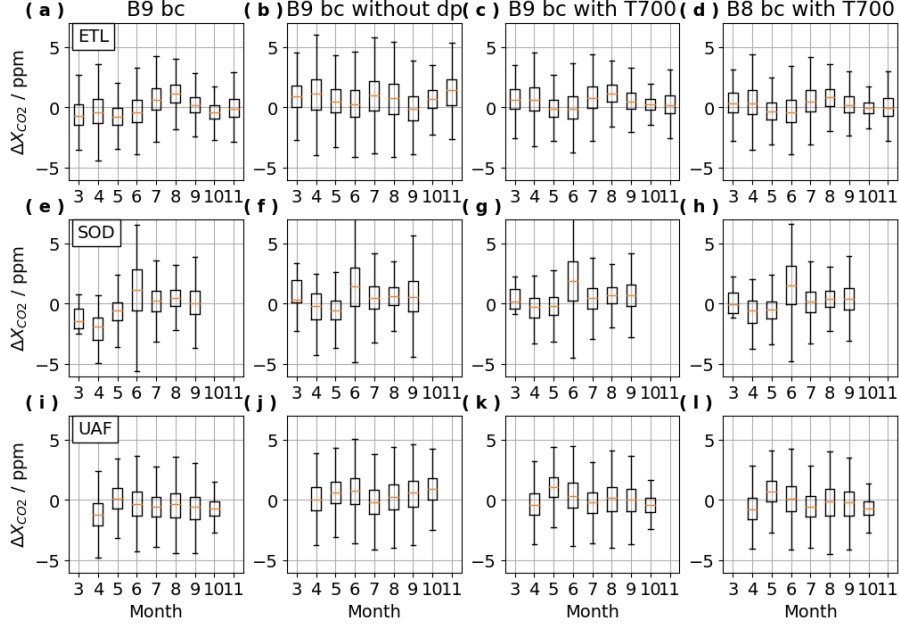

**Figure 15.** Box-plots of distributions of monthly biases at each Boreal Forest site, and filtered by Boreal QC, given the standard B9 bc (Eq. 4), given the B9 bc without the dpfrac term, given the B9 abc (Eq. 6), or given the B8 abc (Eq. 8). Note that bias is defined as ($\Delta X_{CO2} \equiv$ (OCO-2 retrieval)$_i - (\tilde{X}_{\mathrm{NNG}})_i$) for each coincident OCO-2 sounding.

uncertainty in seasonal cycle parameters estimated using OCO-2 retrievals over the Boreal Forest. However, some month-to-month variability persists with any of the bias corrections applied in this paper and it is still important to continue to explore other contributions to seasonal variability in OCO-2 bias, such as the choice of QC, coincidence criteria, or processing of



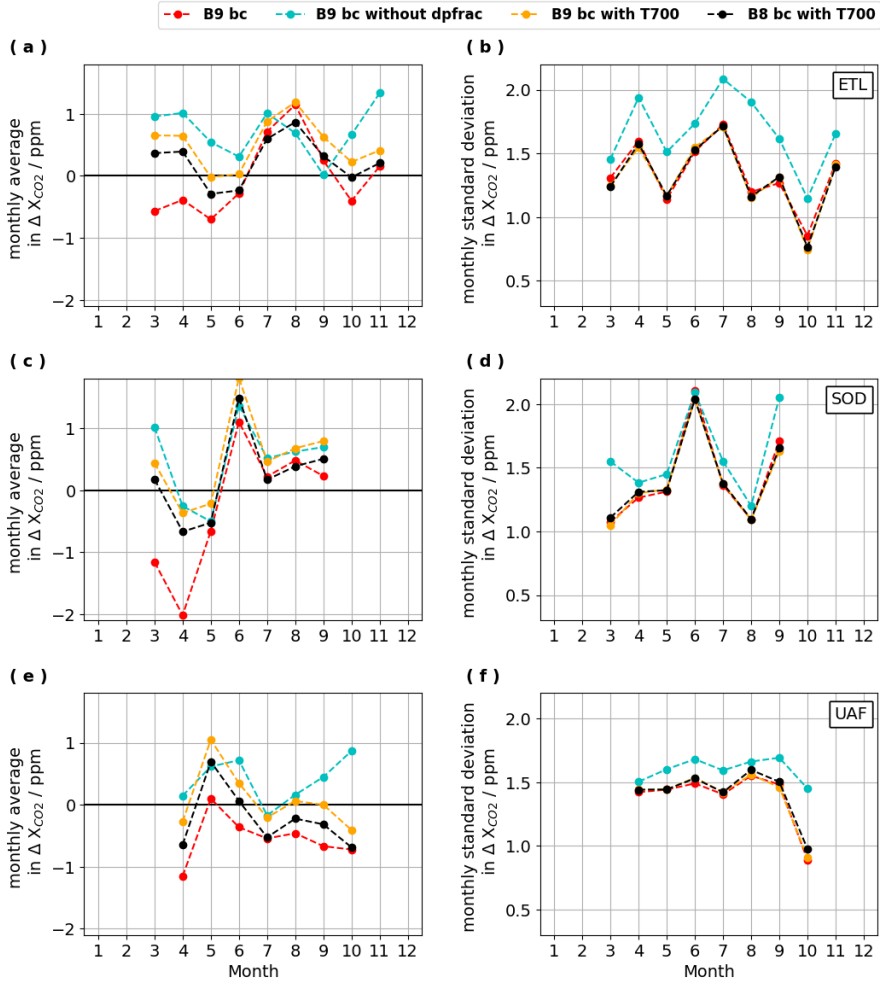

**Figure 16.** Monthly average bias (left) and standard deviation in biases (right) of coincident OCO-2 soundings at each of the three Boreal Forest sites with Boreal QC filtering and given each bias correction modification: the standard B9 bc (Eq. 4), the B9 bc without the dpfrac term, B9 abc with a term for temperature at 700hPa (T700; Eq. 6), and B8 bc with a term for T700 (Eq. 8).

ground-based data. In this vein, the following sections consider differences in monthly average bias and standard deviation in bias between Boreal QC and B9 QC. Then we explore how limiting coincidence by mid-tropospheric temperature, or changing the ground-based reference from NNG to an average of ground-based retrievals at a restricted range of solar zenith angles affects monthly bias distributions. We go on to consider a number of QC parameters that exhibit seasonal behavior and their
5   potential role in seasonally dependent biases at Boreal Forest sites.



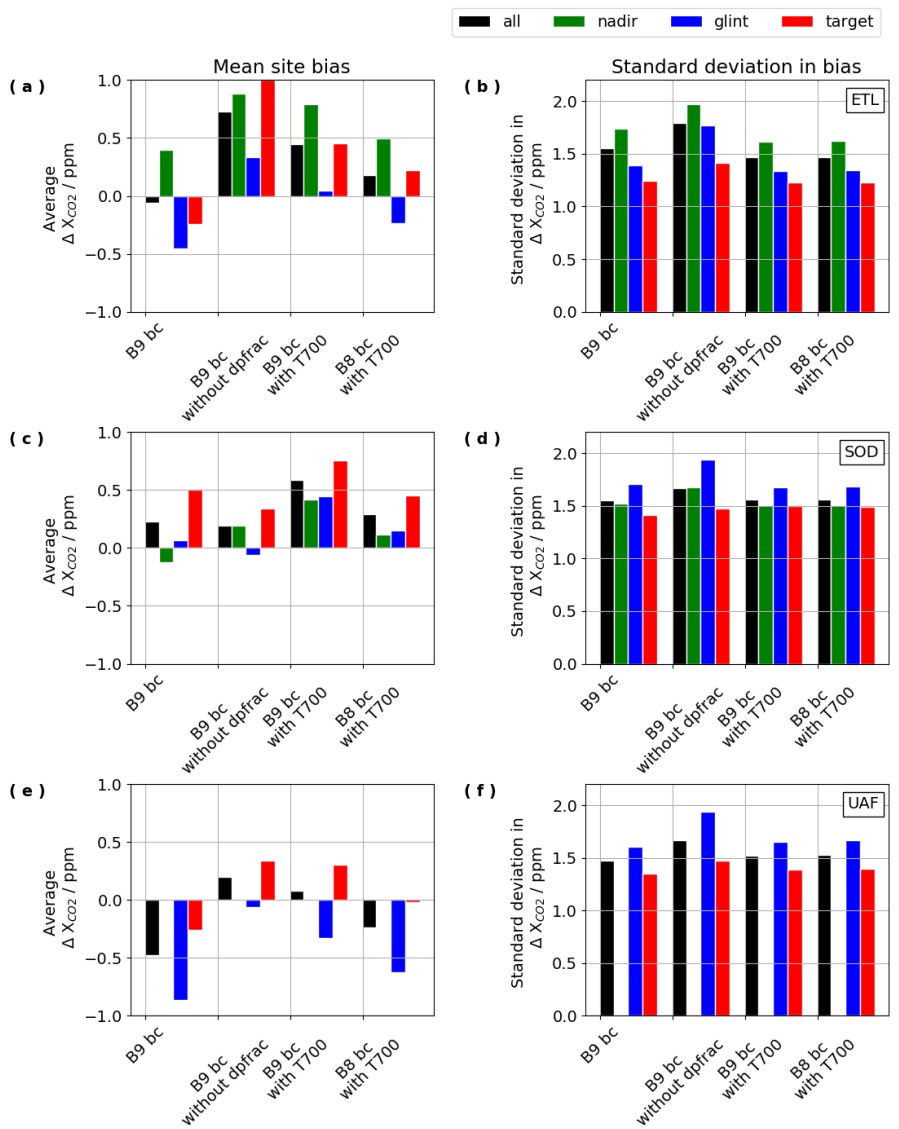

**Figure 17.** Average bias and standard deviation in bias, sorted by viewing geometry and bias correction modification for all coincident soundings at each of the three Boreal forest sites with Boreal QC. Note that there are no coincident nadir soundings for Fairbanks due to the satellites operational design, which favors glint observations in orbits primarily over oceans.

## 4.1 Potential contribution to seasonal bias from QC method

Overall, the B8 QC is the most conservative set of QC filters, the B9 QC allows for more relaxed thresholds in the QC parameters, and the Boreal QC is the most permissive set of QC filters. It was observed in Fig. 9 that the total standard deviations in biases for all coincident soundings at East Trout Lake and Sodankylä gradually increase from $\sim 1.3$ ppm to $\sim 1.4$





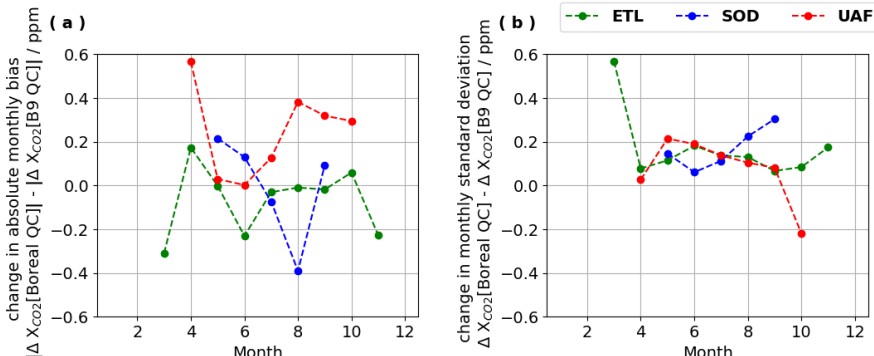

**Figure 18.** Differences between average monthly biases (left) or standard deviation in monthly biases (right) when comparing Boreal QC to B9 QC. Note that Fig. 10 shows typical average monthly bias ranges from -1 ppm to 1 ppm.

ppm to $\sim 1.6$ ppm ascending with the increase in throughput obtained from relaxing QC filters (B8 QC < B9 QC < Boreal QC). A similar trend at Fairbanks is reflected by an increase from $\sim 1.1$ ppm to $\sim 1.3$ ppm to $\sim 1.4$ ppm. Figure 10 also demonstrates this increase in standard deviation with different QC filters, but there does not appear to be a seasonal trend in the monthly standard deviation in biases at East Trout Lake and Fairbanks. The anomalously high standard deviation in biases

in June at Sodankylä remains to be reconciled, and represents a potential complication that would perpetuate mid-summer uncertainty even if some method of correcting seasonal trends in monthly bias were devised and implemented. Additionally, there is substantial increase in standard deviation in biases at East Trout Lake in March with Boreal QC compared to the B9 QC. While this increase in standard deviation is concerning, the availability of OCO-2 retrievals in the Boreal Forest in March remains insufficient for a representative sample of northern regions, and is not likely to be included in seasonal studies of the

Boreal Forest at this time.

The largest difference in the absolute values of monthly bias between Boreal QC and B9 QC is 0.56 ppm in April at Fairbanks (see Fig. 18 panel (a)). Boreal QC also results in a 0.18 ppm larger absolute bias than B9 QC in April at East Trout Lake. In July through October Boreal QC results in monthly biases at Fairbanks that are 0.1 ppm to 0.4 ppm larger than with B9 QC, while in May and June there is no change in average monthly biases between the two QC methods. At East Trout Lake and Sodankylä

Boreal QC produces some monthly biases that are smaller by up to 0.4 ppm than B9 QC. Despite some increases in monthly biases with Boreal QC relative to B9 QC, it is clear from Fig. 18 that the modifications in QC filters do not always result in larger monthly biases, and the effects should be weighed against the potential advantages of increasing passable retrievals and spatial coverage. We conclude that differences between Boreal QC and B9 QC are not likely to be a major source of seasonal variability in bias because seasonal dependence is observed with both QC methods in Fig. 10 and Fig. 11.

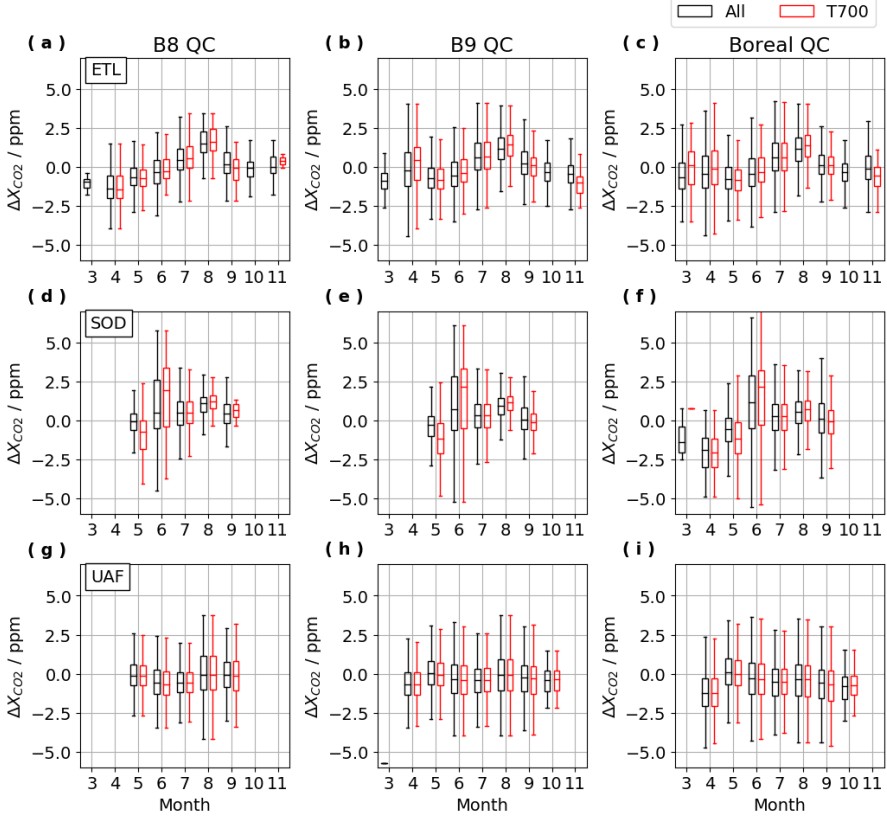

**Figure 19.** Monthly bias distributions for coincident OCO-2 retrievals filtered with Boreal QC comparing the full set of coincident retrievals (black), alongside a subset with the temperature at 700 hPa (T700) in the OCO-2 retrieval equal to $\pm 1K$ of T700 from NCEP reanalysis at the ground site (red).

## 4.2 Potential contribution to seasonal bias from proximity bias

Another suspected source of bias, which may or may not have a seasonal component, is proximity bias. Assuming that the ground-based reference measurements are representative of the full coincidence region depends on a certain amount of regional homogeneity in $CO_2$ columns, and spatial $CO_2$ fields may not meet this criteria during seasonal transitions in spring and autumn. One method applied to GOSAT data by Wunch et al. (2011b) and based on modeling results by Keppel-Aleks et al. (2011) will isolate satellite soundings in the coincidence region that are likely to represent the same atmospheric plume observed at the ground site by choosing soundings with a mid-tropospheric temperature, at 700 hPa (T700), close to that above the ground site. Figure 19 shows monthly bias distributions for the full set of coincident retrievals alongside the monthly bias distributions of a subset of coincident retrievals that have retrieved T700 within $\pm 1K$ of T700 in the daily NCEP reanalysis results for the ground site. Results in Fig. 19 suggest that using T700 to screen coincident retrievals yields little to no observ-





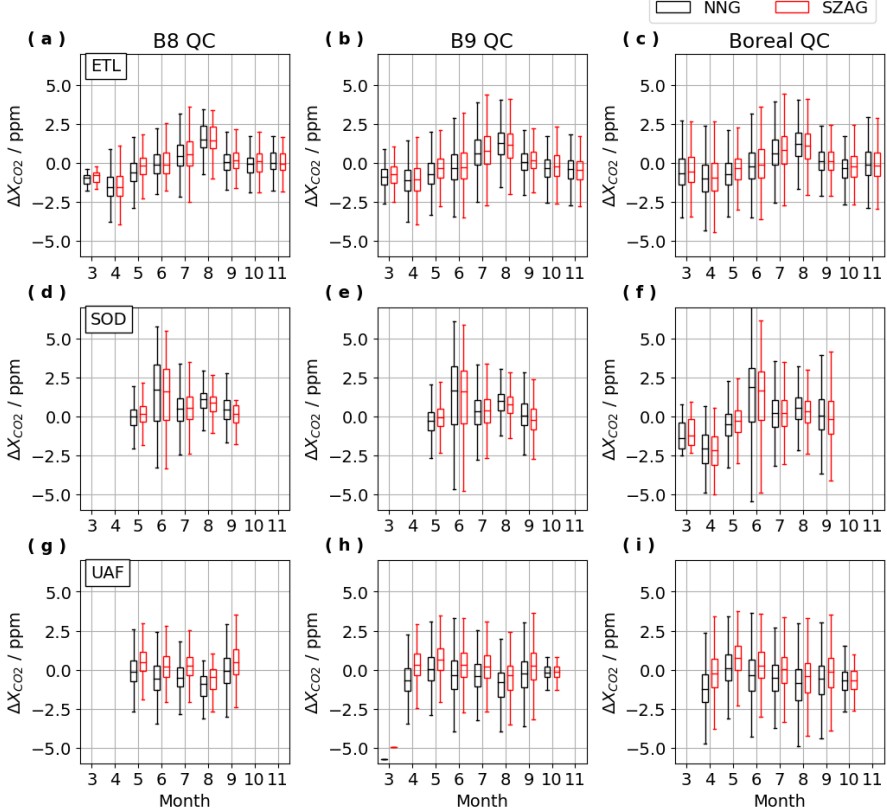

**Figure 20.** Monthly bias distributions for coincident OCO-2 retrievals filtered with Boreal QC and referenced to NNG (black), alongside monthly bias distributions for coincident OCO-2 retrievals with Boreal QC and referenced to SZAG (red).

able improvements in seasonality of biases overall. At East Trout Lake the T700 coincidence criteria does reduce the absolute monthly bias in March and April with Boreal QC, but in a number of other cases T700 screening results in a slight increase in absolute monthly biases for spring and autumn months.

### 4.3 Potential contribution to seasonal bias from ground-based instrument airmass dependence

5   Seasonal biases in NNG data may arise from airmass dependence of ground-based retrievals, particularly in high latitude regions (Wunch et al., 2011a). However, Fig. 20 shows that when the daily ground-based reference is defined as the daily average of retrieved $X_{CO2}$ with apparent sza between 65° and 70° (SZAG) instead of using NNG, there is nearly no observable change in the monthly bias distributions at the two TCCON sites. At Fairbanks, the change in ground-based reference to SZAG results in positive shift in almost all $\Delta X_{CO2}$ values, which for some months corresponds to a reduction in the size of biases

10  and for other months corresponds to an increase in the size of biases. The results at Fairbanks may suggest that sza dependence in EM27/SUN observations requires further study. Seasonal variability in biases persists at all three sites and is largely the





same regardless of whether the ground-based reference is NNG or SZAG. These results suggest that airmass dependence of the ground-based instrument is not likely to be a dominant source of seasonal variability in OCO-2 biases.

### 4.4 Seasonal variability in co2_ratio ($X_{CO2}$[2.06 $\mu$m]:$X_{CO2}$[1.61 $\mu$m])

The co2_ratio refers to the ratio of $X_{CO2}$ retrieved by the 2.06 $\mu$m band to that retrieved by the 1.61 $\mu$m band. Recall that
Wiscombe and Warren (1980) measured low as well as differing reflectance for snow in the 1.61 $\mu$m and 2.06 $\mu$m bands. Systematic departure from unity in the co2_ratio could result from spectroscopic inaccuracies in either band that are characteristic of the instrument or the line-list used in the retrieval algorithm. Anomalous departures from unity in the co2_ratio can arise from low signal to noise ratio in either or both $CO_2$ bands, which can be due to cloud and aerosol interference or the low reflectivity of snow and ice covered surfaces (Crisp et al., 2012). Patchy snow cover or vegetation protruding through the snow may
also cause discrepancies in signal intensity between the weak and strong $CO_2$ bands as a result of variable surface reflectivity in the satellite field of view. In all months at mid-latitudes and in May through October at high latitudes, terrestrial retrievals have a systematic departure from unity in the median co2_ratio, with the data approximately normally distributed around 1.012 (see Fig. 21). There is an even greater departure from unity in the co2_ratio for high latitude retrievals in the winter months, November through April, with the data approximately normally distributed around 1.020. Figure 21 demonstrates that there is
seasonal variation in the median and distribution of co2_ratio at latitudes north of 50°N that is not observed at latitudes from 10°N to 50°N. This monthly difference in the distribution of retrieved co2_ratio at high latitudes may be a symptom of the effects of snow albedo or it may be attributable to some other factor, but it warrants some attention because it may be associated with radiative transfer effects that contribute to negative biases in spring at the Boreal Forest sites.

### 4.5 Total column water vapor (tcwv), bias, and temperature dependence

The parameter tcwv refers to total column water vapor, which is calculated as the product of a scaling factor determined by the full physics retrieval and the a priori tcwv from the European Centre for Medium-Range Weather Forecasts (ECMWF). Atmospheric water vapor is expected to be seasonal and the seasonality of tcwv at the three Boreal Forest sites is illustrated in the box-plots in Fig. 22. Large amounts of atmospheric water vapor can suggest that there may be more cloud cover degrading the quality of both satellite-based and ground-based measurements. Even in the absence of clouds, water vapor is a strong
infrared absorber in all three bands used by OCO-2, and water vapor is identified in Boesch et al. (2019) as the most important absorbing gas interfering with line fitting in OCO-2 retrievals. In selecting QC filters for the Boreal QC, large negative biases (OCO-2 retrievals reporting lower values of $X_{CO2}$ than NNG) were correlated to low tcwv, prompting the introduction of quality thresholds for tcwv in the Boreal QC (see Fig. 23 and Table 1). Figure 24 shows the additional retrievals cut by the lower bound on tcwv at 3 kg m$^{-2}$ (data left of the black dashed line) in the Boreal QC which are not cut by other QC filters,
and an overall downward trend persists in these removed data. One possible explanation is path-shortening resulting from atmospheric scattering, which could result in retrieved spectral radiance that has failed to penetrate atmospheric layers near the surface. This would cause all retrieved gases to be underestimated, so that total column water vapor and the total $CO_2$ column are both erroneously low. However, Fig. 25 shows that while path-shortening may explain some instances of negative biases and

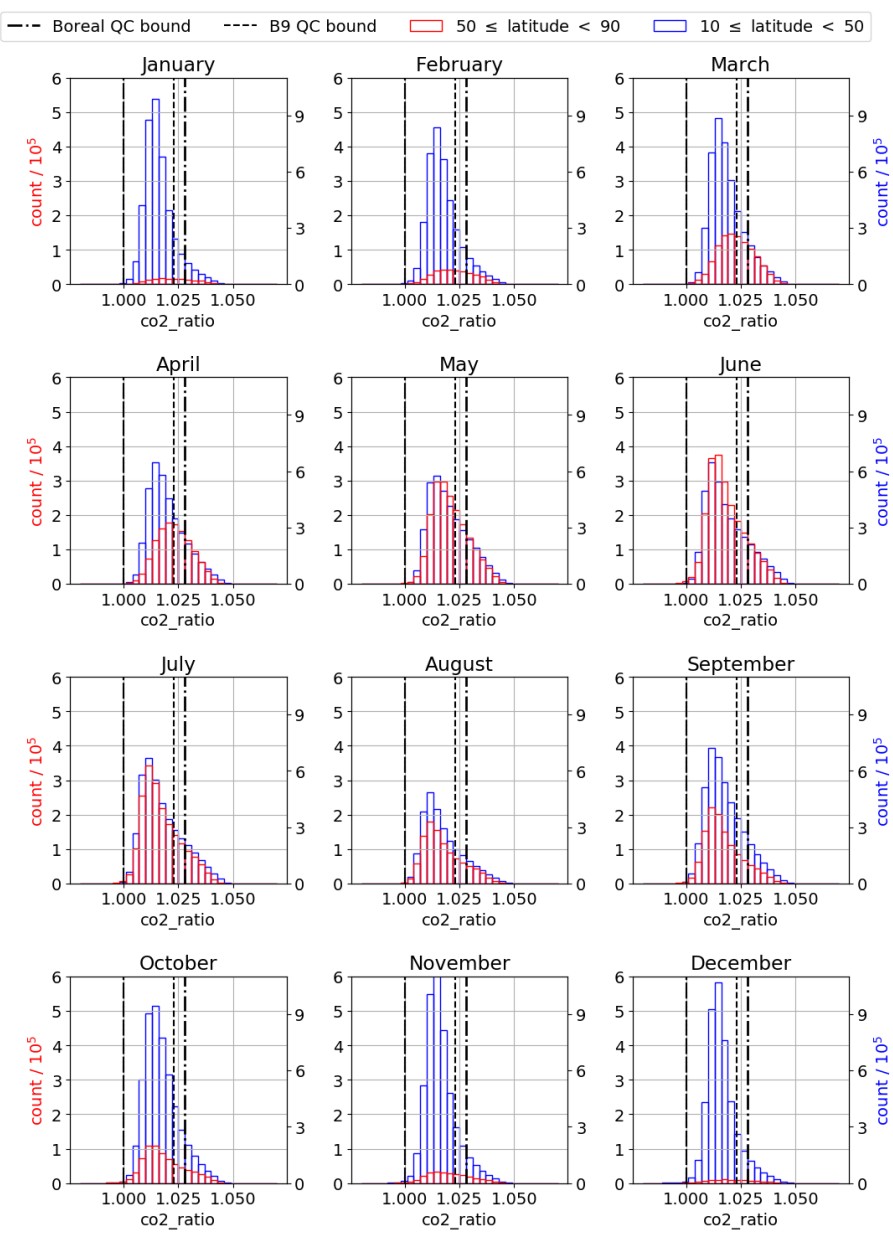

**Figure 21.** Monthly histograms of the ratio in single band retrievals of $CO_2$ (co2_ratio) for all unfiltered OCO-2 retrievals over land (land_fraction=100) split into two latitude bands, $10°N$ to $50°N$ and $50°N$ to $90°N$.

low tcwv, the relationship persists between a priori tcwv from ECMWF reanalysis and negative $X_{CO2}$ biases. Because water vapor is a strong infrared absorber, it would be reasonable to expect retrieval errors when tcwv is high, but low atmospheric water vapor is also associated with cold fronts and snow cover. Figure 26 illustrates the relationship between tcwv and mid-



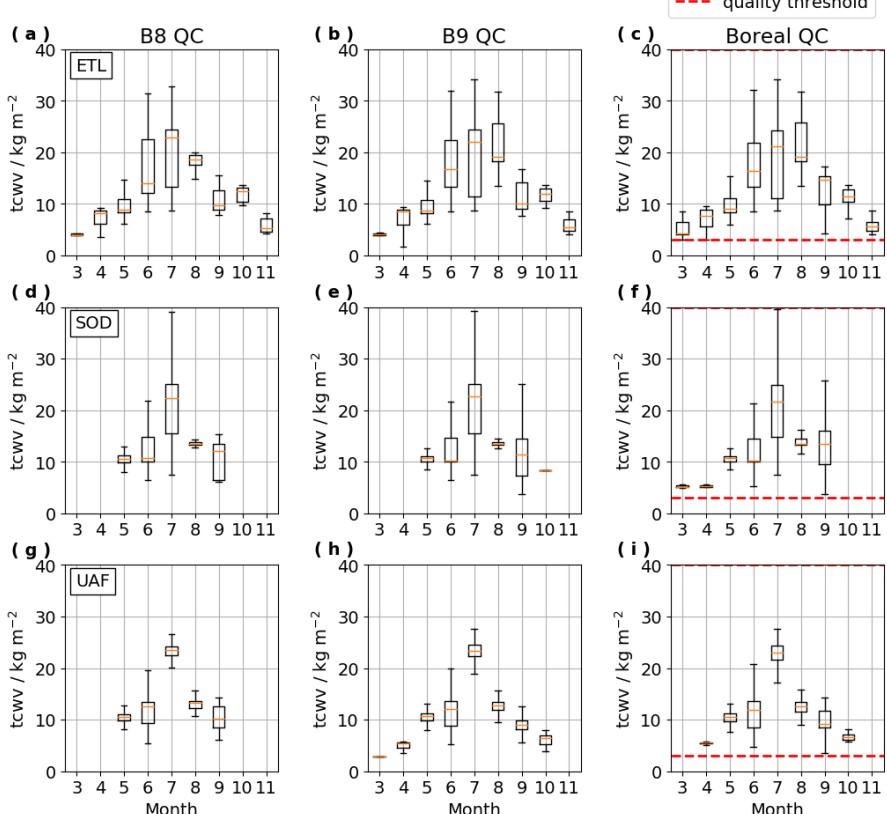

**Figure 22.** Seasonal box-plots of retrieved tcwv (total column water vapor) at (top) East Trout Lake, (center) Sodankylä, and (bottom) Fairbanks given each of the three quality control methods.

tropospheric temperature (T700), at 700 hPa, in Boreal Forest coincident OCO-2 retrievals. There is a distinct maximum for tcwv at a given atmospheric temperature that is defined by the condensation temperature of water, and Fig. 26 shows that most of the retrievals with tcwv below 3 kg m$^{-2}$ are also those with mid-tropospheric temperature (T700) below approximately 250 K. Therefore, it is reasonable to conclude that negative OCO-2 biases are also occurring at low temperatures, which is demonstrated by the correlations between $\Delta X_{CO2}$ and T700 in Fig. 27.

## 4.6 Seasonal variability and temperature dependence in retrieved surface pressure bias

The dp_o2a and dp_sco2 variables are the residuals of retrieved and a priori surface pressure at the pointing locations of the O$_2$A and strong CO$_2$ bands, respectively. These two retrieval parameters were first included in B9 following the discovery of a pointing error that caused systematic inaccuracies in retrieved surface pressure (Kiel et al., 2019). Before the release of ACOS B9 only a single dp variable (the difference between retrieved and a priori surface pressure from GEOS5-FP-IT) was used as a quality control and bias correction parameter. In the analysis by Kiel et al. (2019) an additional parameterization of

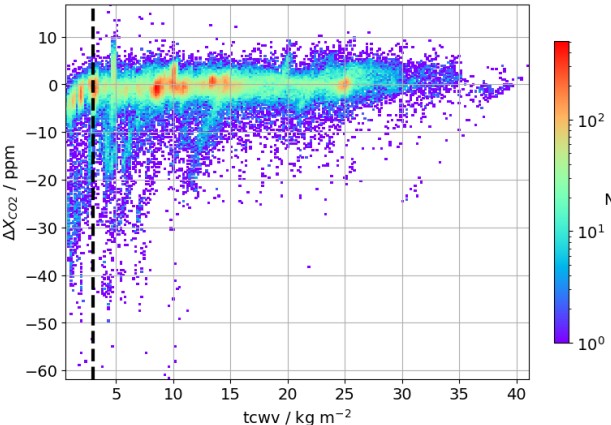

**Figure 23.** Bias in coincident retrievals of $X_{CO2}$ ($\Delta X_{CO2} \equiv$ OCO-2 - NNG) with no QC filtering against retrieved tcwv (total column water vapor), plotted as a density map. The lower bound placed on tcwv in the Boreal QC (3 kg m$^{-2}$) is shown as a black, dashed vertical line.

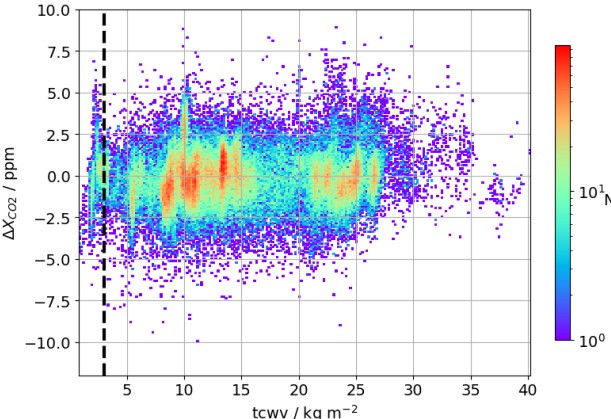

**Figure 24.** Bias in coincident retrievals of $X_{CO2}$ ($\Delta X_{CO2} \equiv$ OCO-2 - NNG) with all Boreal QC filtering except the bound on total column water vapor (tcwv) against retrieved tcwv, plotted as a density map. The lower bound placed on tcwv in the Boreal QC (3 kg m$^{-2}$) is shown as a black, dashed vertical line.

surface pressure residuals (dpfrac) was introduced for use in the OCO-2 B9 bc. The inclusion of dpfrac and dp in the OCO-2 bias correction is not the only reason that surface pressure residuals are important, accurate surface pressure measurements are essential for calculating $X_{CO2}$ which is defined as the ratio of the total CO$_2$ column to the total column of dry air. These terms are essential components of quality control and bias correction methods because even small inaccuracies can translate

5   to unacceptable errors in $X_{CO2}$. While the effects of removing the dp term from the bias correction are considered in section 3.5, it is probably inadvisable to remove this term entirely from bias correction or to loosen quality thresholds on dp variables without careful consideration of the impacts on $X_{CO2}$. Furthermore, in attributing the causes and effects of trends in surface



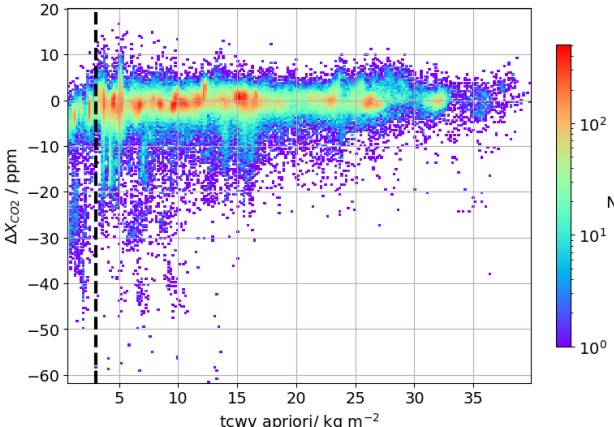

**Figure 25.** Bias in coincident retrievals of $X_{CO2}$ ($\Delta X_{CO2} \equiv$ OCO-2 - NNG) with no QC filtering against a priori tcwv (total column water vapor), plotted as a density map. A priori tcwv is defined by ECMWF reanalysis data. The lower bound placed on tcwv in the Boreal QC (3 kg m$^{-2}$) is shown as a black, dashed vertical line.

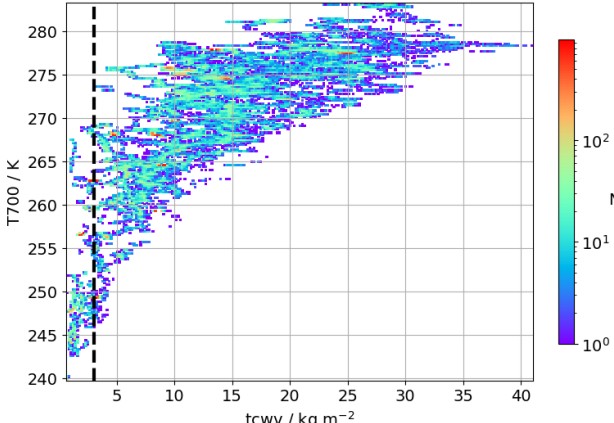

**Figure 26.** Mid-tropospheric temperature (T700 $\equiv$ temperature at 700 hPa) as a function of total column water vapor (tcwv) for all coincident retrievals without QC filters.

pressure residuals there may be many competing factors. The seasonal box-plots in Fig. 28 show that there is a seasonality in all four of the variants on surface pressure residuals (dpfrac, dp_o2a, dp_sco2, and dp) at the three Boreal Forest sites that are similar to the seasonality in bias corrected $\Delta X_{CO2}$ (compare to Fig. 11). Similar seasonality may be a result of multiple seasonal parameters that equally effect dp and $X_{CO2}$. Figures 13 and 14 show that both dpfrac and dp also exhibit linear dependence on T700 with greater linearity than the correlations between $\Delta X_{CO2}$ and T700 given either B9 QC or Boreal QC in Fig. 27 (panels (c) and (d)). Not only is temperature clearly seasonal and correlated to other seasonal parameters, but rates and directions of atmospheric transport are also seasonal and T700 has been found to link plumes in the free-troposphere

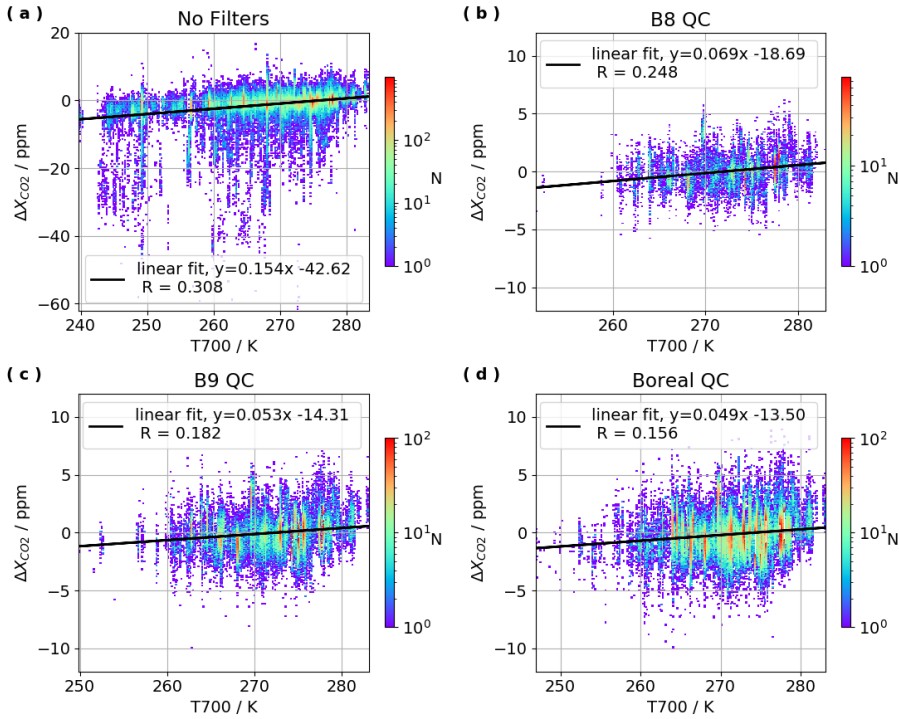

**Figure 27.** Bias in coincident retrievals of $X_{CO2}$ ($\Delta X_{CO2} \equiv$ OCO-2 - NNG) as a function of mid-tropospheric temperature (T700 $\equiv$ temperature at 700 hPa) for all data without QC filters, and each of the three QC methods presented in this paper.

(Keppel-Aleks et al., 2011). Kiel et al. (2019) show that systematic biases in dp are characterized by a positive trend close to the equator and a negative trend at higher southern and northern latitudes, and we claim that this could also be a manifestation of temperature dependence.

## 5 Conclusions

5 Through ILS testing of EM27/SUN spectrometers used in Alaska, regular comparisons between multiple EM27/SUN spectrometers, and comparisons of EM27/SUN spectrometers with TCCON spectrometers we established the relative equivalence of EM27/SUN and TCCON observations as ground-based reference for OCO-2 validation (see supplemental materials section 1). With the application of multiplicative corrections, EM27/SUN FTS measurements in Fairbanks were compared to OCO-2 and yield similar magnitudes in OCO-2 biases as the TCCON sites considered in this study, with less seasonal variability in

10 biases. While there were many challenges with data availability at high latitudes under the B8 QC, both the B9 QC and the Boreal QC offer a two to three fold increase in passable retrievals from OCO-2 Lite files (OCO-2 Science Team/Michael Gunson, Annmarie Eldering, 2018) without major sacrifices in data quality. Total average biases for all sites, viewing modes, and quality control methods were within $\pm 1$ ppm (see Fig. 9). In particular, the Boreal QC allows for nearly twice as many terrestrial





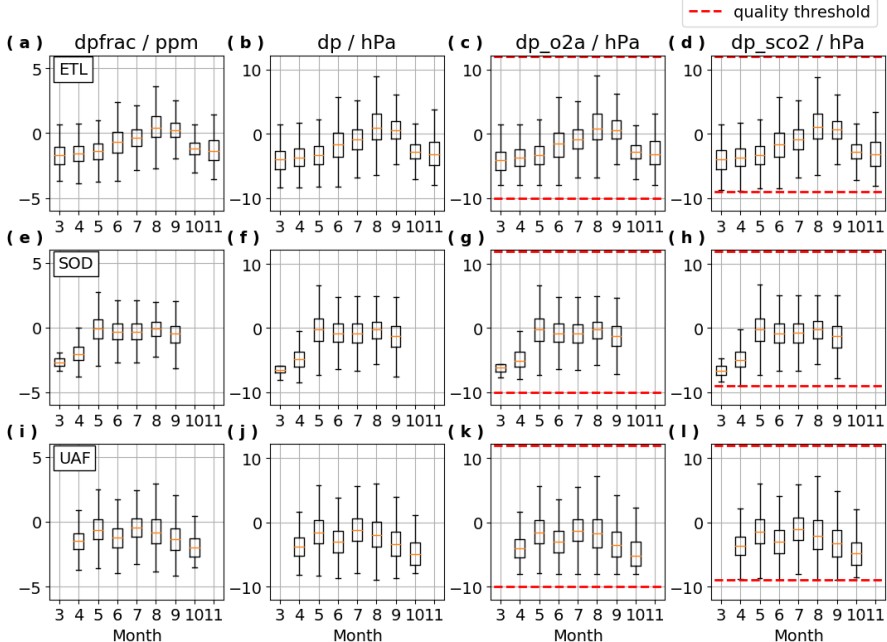

**Figure 28.** Seasonal box-plots of dpfrac, dp, dp_o2a, and dp_sco2 at (top) East Trout Lake, (center) Sodankylä, and (bottom) Fairbanks given Boreal QC filters.

OCO-2 retrievals north of 50°N latitude in the months of May, August, and September relative to the B9 QC, while resulting in no distinct increase in the total average bias and less than 0.3 ppm increase in total standard deviation of biases for coincident retrievals at Boreal Forest sites (see section 3.3.2 and Fig. 9). With the exception of an anomalously large negative April bias at Sodankylä, seasonal variability in monthly biases at these three Boreal Forest sites is mostly confined between -1 ppm and +1

5   ppm, which equates to the 2 ppm maximum monthly standard deviation of biases in June at Sodankylä (when data availability is most abundant, see Fig. 10). Even a slight seasonal trend can significantly impact the analysis of seasonal cycle parameters, so it is important that valid methods for reducing the seasonal dependence in OCO-2 biases are identified. The OCO-2 bias correction seems to introduce some seasonality in OCO-2 bias through the inclusion of a dp (the difference between retrieved and a priori surface pressures) bias correction term. We propose two alternative OCO-2 bias corrections in Eq. 6 and Eq. 8 that

10   correct for temperature dependence in dpfrac and dp, respectively, based on linear regressions shown in Fig. 13 and Fig. 14. It may be important to note that these alternative bias corrections are specifically tailored to high latitude OCO-2 B9 retrievals over land with Boreal QC. Of these two alternative bias corrections, the B8 abc in Eq. 8 appears to be more effective in reducing seasonal variability without substantial increases in average biases in any viewing modes or increases in monthly standard deviations in biases at Boreal Forest sites. The choice of B9 QC or Boreal QC were not found to be a clear source of seasonal

15   dependence in monthly OCO-2 bias in the Boreal Forest, nor were the effects of proximity bias or airmass dependence found to be important contributors to seasonal variability in biases. Several sounding retrieval parameters that have been used as QC





filters were found to exhibit seasonal variability at these Boreal Forest sites, including the ratio of single band retrievals of $CO_2$ (co2_ratio), total column water vapor (tcwv), and the differences between retrieved and a priori surface pressures (dp, dp_o2a, dp_sco2). These parameters may contribute to seasonal variability in biases by impacting data selection in the quality filtering process or they may be indicative of seasonal behavior at high latitudes that is not fully addressed in the retrieval al-

gorithm. In particular, low tropospheric temperatures, or some other parameter that may be correlated to temperature, appears to be one of the primary contributors to seasonal dependence in OCO-2 bias at high latitudes. While the specific choices for QC parameters in the Boreal QC method, proposed here, may still be a subject for consideration and debate, this analysis has shown that it is possible to modify quality controls tailored to a specific region and substantially increase the quantity of usable OCO-2 retrievals with only minor sacrifices in data quality. Furthermore, Boreal QC coupled with an alternative bias correction

that accounts for temperature dependence (Eq. 8) may yield sufficiently stable results for application in preliminary studies of Boreal Forest seasonal cycles of $X_{CO2}$ across longitudes. It is also possible that improvements in spectroscopic modeling in future versions of the ACOS retrieval algorithm would reduce or remove temperature dependence in surface pressure bias.

## Appendix A: Quality control histograms

*Author contributions.* Nicole Jacobs composed this manuscript and conducted the analysis under the supervision of William R. Simpson.

Debra Wunch contributed data from the East Trout Lake TCCON site, as well as guidance in and thorough evaluations of methods and interpretations in the manuscript. Christopher W. O'Dell offered guidance and instructions in methods of evaluating OCO-2 satellite retrieval parameters, bias corrections, and quality controls. Gregory B. Osterman organized targeted satellite overpasses with OCO-2 over Fairbanks and generally oversees satellite validation efforts for OCO-2. Frank Hase, Thomas Blumenstock, Qiansi Tu, Matthias Frey, Manvendra K. Dubey, and Harrison A. Parker all contributed to data collection with the EM27/SUNs in Fairbanks, including instrument evaluations,

maintenance, and establishing long-term operations in Fairbanks. Harrison A. Parker has also acted as local host during calibration to the Caltech TCCON. Rigel Kivi and Pauli Heikkinen operate the TCCON station at Sodankylä, and they provided data and insights into unique aspects of high latitude ground-based measurements of $X_{CO2}$.

*Competing interests.* The authors declare that they have no conflict of interest.

*Acknowledgements.* Thanks to Paul Wennberg, Coleen Roehl, and colleagues at Caltech for operation of the Caltech TCCON and support during side-by-side observations with the LANL EM27/SUN. The Simpson Lab at UAF acknowledges the Alaska Space Grant Graduate





Fellowship and OCO Science Team Grant (NNH17ZDA001N-OCO2) for support. KIT acknowledges support by the ACROSS research infrastructure of the Helmholtz Association of German Research Centres (HGF) and support by the Helmholtz Association in the framework of MOSES (Modular Observation Solutions for Earth Systems). LANL acknowledges NASA CMS and LDRD programs for support.



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



**Figure A1.** Quality control histograms in the same style as those presented by O'Dell et al. (2018), presented for QC parameters used in the selection of Boreal QC and considering only retrievals coincident to the Boreal Forest sites in this paper. Also shown are biases in $X_{CO2}$ with raw OCO-2 retrieved $X_{CO2}$, biases in $X_{CO2}$ with OCO-2 retrieved $X_{CO2}$ bias corrected to TCCON, standard deviation in OCO-2 retrieved $X_{CO2}$ with the bias correction to TCCON and the quality control thresholds selected in the Boreal QC.