# Peer review of "Quality controls, bias, and seasonality of CO2 columns in the Boreal Forest with OCO-2, TCCON, and EM27/SUN measurements"

_Atmospheric Measurement Techniques, 2019_

## Referee Comment (RC1) · Anonymous Referee #1 · 23 Apr 2020

Manuscript "Quality controls, bias, and seasonality of CO2 columns in the Boreal Forest with OCO-2, TCCON, and EM27/SUN measurements" from Jacobs et al. and submitted for publication in Atmos. Meas. Tech. covers and important topic, namely atmospheric CO2 observations at high(er) northern latitudes. The authors study to what extent the currently quite strict quality filtering of the OCO-2 XCO2 can be relaxed / modified to obtain more data at high latitudes and what the impact of this is on data quality. They also address related aspects such as bias correction and they make specific proposals on how to improve. The paper is very well written, contains new interesting and important material and is appropriate for Atmos. Meas. Tech. I therefore strongly recommend publication after the minor aspects listed below have

[Figure]

been considered by the authors.

Specific comments:

Abstract, page 2, line 1 following: Concerning the sentence "... seasonal variability in biases was observed, and this variability was more pronounced at the TCCON sites than when comparing to EM27/SUN observations in Fairbanks." Is that a robust finding, i.e., does this suggest that EM27/SUN is better than TCCON?

Page 3, line 12 following: Concerning the discussion of the trend in the $CO_2$ seasonal cycle amplitude: Please consider also this recent publication: Yin et al., 2018, Changes in the Response of the Northern Hemisphere Carbon Uptake to Temperature Over the Last Three Decades, https://agupubs.onlinelibrary.wiley.com/doi/abs/10.1029/2018GL077316

Page 5, line 28 following: Concerning the suggested new quality control filters for Boreal Forest regions: Are they limited to this region or can they also be used for the global data?

Captions Fig. 3-5: I recommend to write "flagged bad" (e.g. bad with quotes) or equivalent instead of just "flagged". It would then be easier for the readers to understand (if this is correct) that the figures show the number of rejected observations and not the number of accepted observations. Perhaps one may also extend a bit the main text as the figures may wrongly suggest that there are less good data in summer than in winter although the opposite is probably the case (one may also report the fraction (percentage) of rejected pixels per months; does this number depend on season ?).

Figures 5 and 6: Perhaps add also the relative (percentage) increase for each month (just a recommendation; not mandatory as this info is partially provided in Fig. 7).

Typos etc.:

No typos have been identified.

---

## Referee Comment (RC2) · Anonymous Referee #2 · 8 Jun 2020

This paper concerns an important problem encountered when using satellite-based observations to study the atmospheric component of the carbon cycle at high latitudes, namely the relative scarcity of satellite XCO2 data compared with that available over the mid-latitudes and the tropics. The difficulty in making ground-based observations in these regions, owing to harsh conditions and lack of accessibility and infrastructure, adds to the importance of satellite observations when studying the arctic region.

Jacobs et al. address this problem by considering the data quality filtering applied to OCO-2 data, noting that the quality filters applied as standard to the global dataset - with the goal of minimising bias and scatter compared with ground-based validation

measurements made by TCCON sites around the world - may not be the most appropriate for a study focused purely on the Boreal regions. The goal, therefore, is to investigate whether certain quality filters and thresholds can be adjusted to increase the amount of quality filtered data over the Boreal region without significantly compromising the data quality.

In order to investigate the effect of introducing different quality filters on the bias and scatter, ground-based measurements are required to provide baseline XCO2 data to validate against. Data from two TCCON sites (East Trout Lake and Sodankyla) are used here, along with campaign measurements from two EM27/SUN instruments based in Fairbanks, Alaska. These instruments have become more established in satellite validation in recent years, and studies referred to in this paper have demonstrated that their performance is comparable to that of the TCCON stations. The EM27/SUNs are initially calibrated against the Caltech TCCON, and the same retrieval algorithm (GGG2014) is used on both TCCON and EM27/SUN throughout. This step is essential in that it allows the data from all three sites to be regarded as interchangeable, regardless of the instrument used at each site.

Jacobs et al. provide a detailed analysis of how their proposed quality control filter (Boreal QC) performs compared with two filters recommended by the OCO-2 science time (B8 QC and B9 QC), with great care taken to consider the impacts of specific thresholds on the filtered data. The increase in data throughput achievable whilst only introducing minor changes in average bias is an important outcome of this paper. They do however note that the biases exhibit some seasonal variability, mostly independent of the QC method used, which contribute to uncertainty in characterising seasonal cycles of carbon dioxide. It was interesting to see that it is the global bias correction itself that introduces the delta XCO2 seasonality seen in the Boreal data: I think this point could be made clearer by combining Figures 11 and 12 (for example, by plotting the bias-corrected and non-bias-corrected data on the same axes). An alternative bias correction (abc), taking into account the temperature dependence identified as

introducing the seasonality in the bias over the Boreal region, is derived and introduced for the B8 and B9 QC datasets, and shown to reduce (but not completely eliminate) the seasonality in the monthly average bias without affecting the monthly standard deviation.

The final part of the paper considers possible explanations for the differences in monthly average bias and standard deviation in bias between the Boreal and B9 QC filtered data, and looks at whether coincidence criteria or particular QC parameters that exhibit seasonal behaviour have a role in the seasonally dependent biases observed. Where it is concluded that a parameter is not likely to be a dominant source of seasonal variability in OCO-2 biases (proximity bias, ground-based instrument airmass dependence), I suggest moving these sub-sections to an appendix in order to make it more clear which parameters most urgently require further study.

Overall, this paper offers an in-depth consideration of how quality control filters applied globally to OCO-2 XCO2 data may not be the most appropriate for studying specific regions where the number of quality filtered data points becomes prohibitively low. The proposed Boreal QC filter is tested thoroughly in comparison with the B8 and B9 QC filters, and limitations with all three filters related to the seasonality in the XCO2 bias are identified and considered at length. A further outcome of this paper is that it demonstrates how, when supported by side-by-side pre-campaign calibration observations, the EM27/SUN and TCCON calibration data can be considered as equivalent (although the authors do rightly note that possible airmass dependence biases in the EM27/SUN data still require investigation). A final suggestion I have is that the Boreal QC filtered dataset could be made available in its own right, to allow for further study of the seasonality introduced by the bias correction.

I am happy to recommend this paper for publication, with the following suggested minor technical/organisational changes:

Pages 20/22: combine Figures 11 and 12 into a single figure, to emphasise how the

seasonality is introduced by the OCO-2 bias correction.

Pages 23/24/25: in Figures 15/16/17, for consistency with the text, I suggest using the label 'abc' instead of 'bc' when showing the 'with T700' results.

Sections 4.2 and 4.3: I suggest moving these (and their associated figures) to an appendix, so that the discussion section is more focused on the QC parameters which have a greater impact on the seasonal variability in the XCO2 biases.

Appendix A: a paragraph here which briefly describes how the quality control histograms are used to obtain the QC thresholds would be helpful (the authors may still refer to the O'Dell paper here for further detail).

---

## Author Comment (AC1) · 4 Jul 2020

We are grateful to referee #1 for their review of this manuscript and identification of important points where clarification was needed. The referee's endorsement of the paper is appreciated. In the text below, we detail responses to each of the referee's comments. Page and line numbers refer to the originally submitted discussion paper.

referee comment: "Abstract, page 2, line 1 following: Concerning the sentence "...seasonal variability in biases was observed, and this variability was more pronounced at the TCCON sites than when comparing to EM27/SUN observations in Fairbanks." Is that a robust finding,i.e., does this suggest that EM27/SUN is better than TCCON?"

[Figure]

Our study uses only two instruments from the TCCON network and one EM27, all of which are deployed at different locations, so the present work is not sufficient to suggest which instrument is better, and that answer might also depend upon the application. It is possible that the seasonal stability in OCO-2 bias relative to the EM27 at Fairbanks could be due to the EM27 spectral resolution ($\sim$0.5 cm-1), which is much closer to that of OCO-2 ($\sim$0.3 cm-1) than TCCON ($\sim$0.02 cm-1). While the EM27s used in Fairbanks were calibrated against the Caltech TCCON, there is no TCCON in Fairbanks and vertical distributions of $CO_2$ at these two sites are likely to differ. An analysis of OCO-2 data compared to long term EM27 observations at multiple TCCON sites would be necessary to more fully address whether the relative seasonal stability in OCO-2 bias at Fairbanks is due to geography or instrumentation. There are a small number of sites that have published analyses or data from EM27 observations alongside a TCCON for multiple consecutive years, but we are not aware of any that have published results comparing multiple years of EM27 measurements to OCO-2. Sha et al., 2019 (AMTD, doi: 10.5194/amt-2019-371) present thorough comparisons amongst EM27 observations, TCCON observations, and TCCON measurements truncated to a lower resolution, as well as other infrared spectrometers, using a full year of measurements at Sodankyla, Finland. Their results suggest that the EM27 may retrieve higher XCO2 in spring than the TCCON, and this could result in reduced seasonal dependence in bias. Their results also suggest that factors such as temperature and water vapor may play a role in the differences between EM27 and TCCON retrievals of XCO2. In general, the EM27 is a relatively new instrument and has yet to undergo the extensive effort that has been applied to TCCON measurements, so more research is needed. A sentence was added to the abstract and conclusions to clarify this point, and the discussion in section 4 (page 21, line 28) was enhanced with this citation and further discussion.

referee comment: "Page 3, line 12 following: Concerning the discussion of the trend in the CO2 seasonal cycle amplitude: Please consider also this recent publication: Yin et al., 2018, Changes in the Response of the Northern Hemisphere Carbon Uptake to Temperature Over the Last Three Decades, https://agupubs.onlinelibrary.wiley.com/doi/abs/10.1029/2018GL077316"

Thank you for calling attention to this study, which will also be helpful in future research of Boreal Forest seasonal cycles. Citation of Yin et al., 2018, was added at page 3 line 12 and some additional description of their findings was added at the end of the first paragraph in the introduction.

referee comment: "Page 5, line 28 following: Concerning the suggested new quality control filters for Boreal Forest regions: Are they limited to this region or can they also be used for the global data?"

The alternative set of quality controls proposed in this manuscript are specifically validated in the Boreal Forest (the biome covering the majority of land between 50N and 70N latitude) and yield the optimal combination of throughput and data quality when applied to these data. Our analysis only vetted these quality control parameters using ground-based observations from three Boreal Forest sites, so it is unknown how they will affect data quality when applied to regions south of 50N or over the high Arctic (Greenland and places with abundant sea ice). The Boreal QC yields slightly higher standard deviation than the B9 QC, and while this may be a worthwhile sacrifice to increase data throughput at high latitudes, it may not be worthwhile for a region that already has high data throughput with B9 QC. In particular, some aerosol filters were removed in the Boreal QC, which are redundant for these Boreal sites but could be important over urban areas. Stricter bounds placed on the slope of the continuum albedo in the strong CO2 band (albedo_slope_sco2) in the Boreal QC may also cause unexpected cuts in regions outside of what was considered. If one intended to apply the Boreal QC to another region, we recommend a careful comparison to well-established ground-based validation data within the region of interest before making any decision. While there is some effort made to clarify this point within the paper, we also added some language to the end of the introduction (near page 5, line 28) to clarify that the alternative quality filters are designed and vetted for the Boreal Forest.

referee comments: "Captions Fig. 3-5: I recommend to write "flagged bad" (e.g. bad with quotes) or equivalent instead of just "flagged". It would then be easier for the readers to understand(if this is correct) that the figures show the number of rejected observations and not the number of accepted observations. Perhaps one may also extend a bit the main text as the figures may wrongly suggest that there are less good data in summer than in winter although the opposite is probably the case (one may also report the fraction(percentage) of rejected pixels per months; does this number depend on season ?).

Figures 5 and 6: Perhaps add also the relative (percentage) increase for each month(just a recommendation; not mandatory as this info is partially provided in Fig. 7)."

First, captions in Fig. 2, 3, and 4 were modified to read "flagged bad", as suggested, and corresponding references in the text were also changed from "flagged" to "flagged bad". In response to your suggestion about showing fraction (percentage) of rejected pixels, we took what may be a slightly different approach than intended and decided to modify Figures 5, 6, and 7, as well as making other revisions to section 3.2. Figures 5 and 6 were the maps of additional spatial coverage from using Boreal QC instead of B9 QC, and they were changed to show the difference in the number of soundings between Boreal QC and B9 QC in each 1x1 degree geographic grid cell, on a diverging color scale and including small reductions in throughput. These maps are now much more informative and give a more holistic perspective on the changes in throughput from the Boreal QC. As suggested, we changed Fig. 7 to show the fraction of soundings passed by each QC relative to the total number of soundings over land north of 50N rather than showing the absolute number of soundings passed. We believe that this better represents the improvements in data throughput because for some months there are already fewer data points prior to quality filtering as a result of pre-screening done on the OCO-2 Lite file data or from lack of data collection.

---

## Author Comment (AC2) · 4 Jul 2020

We are grateful to referee #2 for constructive suggestions for improvement. The referee's favorable account of the implications of these results for improving satellite data throughput in under-sampled regions and validating EM27 measurements against TCCON are also appreciated and reflect what we had hoped to communicate in this manuscript. In addition, the referee makes a good suggestion to provide a publicly available dataset with Boreal QC filtered OCO-2 data. Major efforts have been put into developing and validating the globally appropriate B9 QC scheme (and subsequent QC releases) for OCO-2 and the Boreal application is a more specific one, so we do not

think it appropriate to release OCO-2 data with the Boreal QC filtering already applied. However, all of the filtering parameters are defined in Table 1 with names that match their corresponding variables in the OCO-2 Lite files, so an interested group could follow the methods described in this manuscript and code a script to apply the filters to data in the OCO-2 Lite files, either independently or by contacting us to collaborate and share code. All of the referee's recommendations for organizational revisions were implemented with the following details. Page and line numbers refer to the originally submitted discussion paper.

referee comment: "Pages 20/22: combine Figures 11 and 12 into a single figure, to emphasize how the seasonality is introduced by the OCO-2 bias correction."

Figures 11 and 12 were combined into side-by-side boxplots of a similar style to previously numbered Fig. 19 and Fig. 20. As an extension of this change, the color-scheme and formatting of all boxplots, including previously numbered Fig. 15, 19, 20, 22, and 28, were changed to improve clarity and aesthetics.

referee comment: "Pages 23/24/25: in Figures 15/16/17, for consistency with the text, I suggest using the label 'abc' instead of 'bc' when showing the 'with T700' results."

Figures 15, 16, and 17 were relabeled for consistency with the text as suggested.

referee comment: "Sections 4.2 and 4.3: I suggest moving these (and their associated figures) to an appendix, so that the discussion section is more focused on the QC parameters which have a greater impact on the seasonal variability in the XCO2 biases."

Sections 4.2 and 4.3 were added as appendices and text in the discussion and conclusions was altered in accordance with this change.

referee comment: "Appendix A: a paragraph here which briefly describes how the quality control histograms are used to obtain the QC thresholds would be helpful (the authors may still refer to the O'Dell paper here for further detail)."

A descriptive paragraph was added to Appendix A describing the QC histograms and giving some brief explanation of how they are used to determine quality control filters.